

# 5d SCFTs and their non-supersymmetric cousins

Mohammad Akhond[1*], Masazumi Honda[2,3†] and Francesco Mignosa[4‡]

**1** Department of Physics, Kyoto University, Kyoto 606-8502, Japan
**2** Yukawa Institute for Theoretical Physics, Kyoto University, Kyoto 606-8502, Japan
**3** Interdisciplinary Theoretical and Mathematical Sciences Program (iTHEM),
Riken, Wako 351-0198, Japan
**4** Department of Physics, Technion, Israel Institute of Technology, Haifa, 32000, Israel

* akhond@gauge.scphys.kyoto-u.ac.jp , † masazumi.honda@yukawa.kyoto-u.ac.jp ,
‡ francescom@campus.technion.ac.il

## Abstract

We consider generalisations of the recently proposed supersymmetry breaking deformation of the 5d rank-1 $E_1$ superconformal field theory to higher rank. We generalise the arguments to theories which admit a mass deformation leading to gauge theories coupled to matter hypermultiplets at low energies. These theories have a richer space of non-supersymmetric deformations, due to the existence of a larger global symmetry. We show that there is a one-to-one correspondence between the non-SUSY deformations of the gauge theory and their $(p,q)$ 5-brane web. We comment on the (in)stability of these deformations both from the gauge theory and the 5-brane web point of view. UV duality plays a key role in our analysis, fixing the effective Chern-Simons level for the background vector multiplets, together with their complete prepotential. We partially classify super-Yang-Mills theories known to enjoy UV dualities which show a phase transition where different phases are separated by a jump of Chern-Simons levels of both a perturbative and an instantonic global symmetry. When this transition can be reached by turning on a non-supersymmetric deformation of the UV superconformal field theory, it can be a good candidate to host a 5d non-supersymmetric CFT. We also discuss consistency of the proposed phase diagram with the 't Hooft anomalies of the models that we analyse.



# 1 Introduction

Superconformal field theories (SCFTs) in five spacetime dimensions have been an active area of research, following their conception in the seminal work by Seiberg [1]. Their existence is heavily reliant on stringy constructions such as brane webs [2], geometric engineering [3], as well as their holographic dual $AdS_6$ solutions [4–7]. For a recent review see [8]. An important component in the aforementioned class of CFTs is the existence of supersymmetry (SUSY). In contrast, non-supersymmetric fixed points are still poorly understood, and their very existence is a matter for debate. Some notable progress was achieved employing the conformal bootstrap and the $\epsilon$-expansion [9–20]. Yet, we still have no conclusive evidence of the existence of these CFTs, not least because non-perturbative methods based on string constructions are not fully under control in this setting.

Recently, soft-SUSY breaking deformations of the $E_1$ SCFT were analyzed in [21] as a possible way to flow from a UV SCFT to a non-SUSY CFT in the IR. Indeed, the $E_1$ SCFT admits a supersymmetric mass deformation, which triggers an RG flow, leading at low energies to an $\mathcal{N}=1$ $SU(2)$ super-Yang-Mills (SYM) theory. The corresponding deformation parameter $\langle\phi_a\rangle$ is the vacuum expectation value (VEV) for the lowest component of the background vector multiplet $\mathcal{V}_I$ for the $SO(3)_I$ symmetry acting on the Higgs branch chiral ring.[1] Selecting the third component $\langle\phi_3\rangle$, we can identify the inverse gauge coupling $\frac{1}{g^2}$ of the IR free gauge theory as $\langle\phi_3\rangle \equiv h \sim \frac{1}{g^2}$. This deformation explicitly breaks the $SO(3)_I$ symmetry to its Cartan subgroup, which at low energies and at the origin of the Coulomb branch (CB) describes the topological $U(1)_I$ symmetry of the gauge theory with conserved current

$$J_I[F] = \frac{1}{8\pi^2} \star \mathrm{Tr} F \wedge F. \tag{1}$$

Further deforming the theory by turning on a VEV $\langle D_a^i\rangle = \langle D_3^3\rangle \equiv d$ for the highest component of $\mathcal{V}_I$ breaks supersymmetry. This follows because this operator carries an index $i$ in the adjoint

---

[1]That the global symmetry of the $E_1$ SCFT is $SO(3)$ rather than $SU(2)$ was first suggested in [22].

of the $SU(2)_R$ R-symmetry, breaking it down to its $U(1)_R$ Cartan subgroup. Both the gaugino and the scalar gaugino acquire a mass proportional to $d/h$ and can be integrated out in the weak coupling regime $|d| \ll h^2$, hence the theory flows to pure SU(2) Yang-Mills. The $E_1$ fixed point admits another mass deformation, leading to pure SU(2) gauge theory at weak coupling, where one turns on the mass parameter with the opposite sign $\hat{h} = -h$. When the mass parameter $|h|$ is small compared with the scale of the Coulomb branch, both theories describe the vicinity of the fixed point and are in this sense referred to as UV-duals. The duality map between the two theories, in addition to the sign flip $h \rightarrow -h$, includes a shift of the Coulomb branch VEV, this generates a $\mathbb{Z}_2$ orbit which is identified with the Weyl group of the $SO(3)_I$ symmetry. This $\mathbb{Z}_2$ action changes both the sign of the Chern-Simons level associated with the background vector field of the $U(1)_I$ global symmetry and of the residual $U(1)_R$ symmetry, which can both be shown to be non-zero in the pure YM phase. Observing the change of sign in the effective Chern-Simons level for the background $U(1)_I$ and $U(1)_R$ symmetries, the authors of [21] showed that a phase transition must separate the two YM phases related by the $\mathbb{Z}_2$ Weyl group action. In particular, in the hypothesis that the whole $U(1)_I \times U(1)_R$ symmetry remains unbroken at the transition point, the change of levels is driven by both perturbative and non-perturbative particles becoming simultaneously massless at the transition point. This is the hallmark of a possible interacting fixed point. However, both the order of the transition and the position of the phase transition line in the $(h, d)$ plane were left undetermined.

This picture was further refined in [23], where the deformation was analyzed in the context of the $(p, q)$ web description of the $E_1$ theory. It is well known that supersymmetric mass deformations of 5d SCFTs correspond to deformations in the plane of the $(p, q)$ 5-brane. In contrast, the SUSY-breaking deformation $d$ was identified with a particular rotation transverse to the plane of the $(p, q)$ 5-brane for the $E_1$ theory.[2] The string theory analysis revealed an instability on the Higgs branch as one turns on the SUSY breaking deformation $d$, which was confirmed by analyzing the effect of the deformation at a generic point of the Higgs branch of the SCFT. The instability induces a tachyon condensation mechanism in string theory, which can be interpreted in field theory terms as a spontaneous breaking of the $U(1)_I$ symmetry. However, the fate of the system after this instability is, in general, difficult to determine, except for the infinite coupling axes, at which the potential appears unbounded and the instability cannot be resolved [23]. The analysis suggests figure (1) as the minimal phase diagram for the $E_1$ SCFT. The $E_1$ fixed point sits at the $SO(3)_I \times SU(2)_R$ symmetric origin of the two-dimensional $(h, d)$ plane. Far along the $(\pm h, 0)$ direction, the theory is described by pure SU(2) SYM which has an $SU(2)_R \times U(1)_I$ global symmetry. The YM regime corresponds to the regime $h^2 \gg |d|$. There is a symmetric region around the $d$-axis where $U(1)_I$ is spontaneously broken. The boundary of this region is marked by the fuzzy cyan curve in figure (1), representing a cartoon of the actual phase transition curve. Determining the shape of this curve is difficult due to our ignorance of the precise form of the tachyon potential in the finite coupling region. The order of the phase transition on this line remains an open question.

Given the rich landscape of 5d SCFTs, it is natural to extend the logic pioneered in [21] to other theories. A step in this direction has already been taken in [26], where a particular non-supersymmetric mass deformation for the $X_{1,N}$ theory [27] was analyzed. In particular, the authors found evidence for a second order phase transition in the large $N$ limit of the theory by analyzing the $(p, q)$ web subject to the non-SUSY and SUSY deformations. Moreover, the complete prepotential of the $X_{1,N}$ theory was recently constructed in [28]. However, there is so far no further field theoretic generalisation of the analysis performed for the $E_1$ theory. The goal of this paper is to bridge this gap by analyzing higher rank generalisations of the $E_1$ fixed point. In particular, we consider SCFTs which admit a gauge theory deformation

---

[2]The analogous deformation in 3d Hanany-Witten setups was analyzed in [24, 25], which led to new non-supersymmetric dualities in 3d gauge theories.

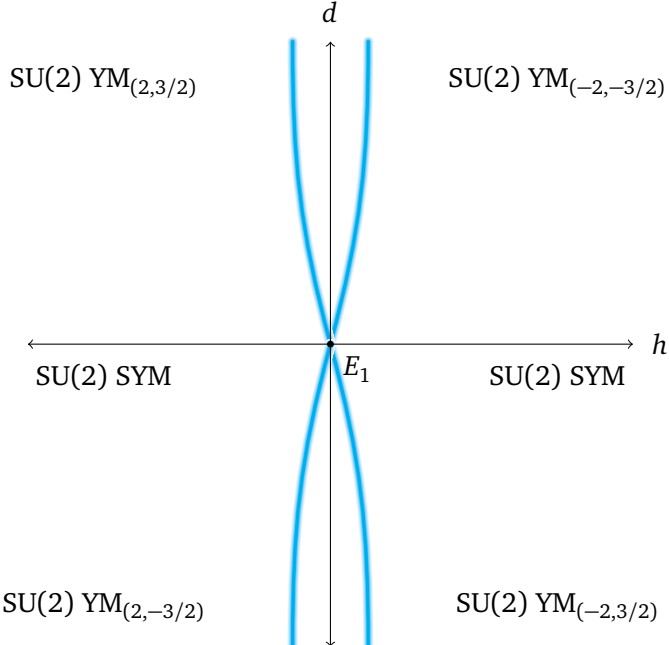

Figure 1: Summary of the phase diagram of the $E_1$ theory. The physics in each of the four quadrants is identical and related by the $\mathbb{Z}_2 \times \mathbb{Z}_2$ Weyl group action of the $SO(3)_I \times SU(2)_R$ symmetry. The fuzzy cyan line separates the symmetry broken phase from the pure YM ones. The exact shape of this line and the order of the phase transition taking place there is still not fully understood.

to a pure SYM theory. A crucial ingredient in determining the background CS level for the instantonic symmetry of the $E_1$ theory was the existence of UV self-duality relating the weakly coupled descriptions of the $E_1$ theory with mass parameter $\pm h$. In string theory, this follows from the S-duality of type IIB. As we will see, in theories with self-duality we will be able to completely fix the CS level associated with the global symmetries and so determine the jump between the two low energy phases obtained by deforming the SCFT point by both the SUSY and the non-SUSY mass deformations. When the SCFT admits two different IR gauge theory descriptions, UV duality will not be able to completely fix the jump of CS levels. In such cases, we will determine the background CS levels up to some integer constant. In both cases, the calculations will be verified by explicitly constructing the complete prepotential of the theory and calculating the CS levels directly from it. We will also discuss consistency of the proposed phase diagram with existing 't Hooft anomalies of the models that we analyse.

The rest of the paper is organized as follows. In section 2, we study the effect of a generic SUSY breaking deformation involving the background vector multiplet of the flavor symmetry of a generic SYM theory coupled to $F$ hypermultiplets. This will allow us to set the stage and understand the weakly coupled phases of the theory obtained after mass deforming the SYM theories. The non-supersymmetric deformations leading to an instability already visible at weak coupling will be discarded, while the ones leading to a stable vacuum will be analyzed in the later sections. In section 3, we generalize the calculation to $SU(N)_N$ theories, calculating their complete prepotential and the corresponding jump of CS levels associated with the $U(1)_I$ global symmetry of the corresponding weakly coupled descriptions. This will be achieved by employing the self-duality enjoyed by these theories. We then generalize the story to the UV fixed point of the dual SU-Sp theories, employing their UV duality, determining the jump of the global symmetry CS levels up to an integer value. In section 4, we will analyze the rank-$N$ $E_1$ SCFT, determining via self-duality the jump of the CS levels and its CFT prepotential. The

description of the SUSY breaking deformation will be verified in detail by looking at its $(p, q)$ web description, which shows interesting complicated dynamics. In section 5, we analyze the case SU(4)$_0$+2AS. This is meaningful, since the theory can be reduced to the Sp(2)+1AS theory through Higgsing and to SU(2)$_2$ if we go at infinite distance on the CB, so it can be seen as a "mother theory" for most of the cases we treated above. We then conclude and give some outlook of our results.

## 2 Weak coupling analysis

In order to determine the phase diagram of SCFTs subject to the non-SUSY and SUSY mass deformations we would like to study, we should analyze their gauge theory descriptions. The gauge theory phase can be reached by flowing in the IR after turning on a SUSY mass deformation. We can then turn on a SUSY breaking mass deformation. This regime is equivalent to the limit $1/g^2 \gg \sqrt{d}$, where $1/g^2$ represents the gauge coupling associated with the gauge theory, which we take to be simple, and $d$ is the non-SUSY mass parameter. In the following, we will in particular focus on non-SUSY mass deformations involving the global symmetry current multiplet, although other types of non-SUSY deformations are available in principle [29]. The basic reason underlying this choice comes from the possibility of geometrically realising these deformations in the $(p, q)$ web. Since they break both the global and the R-symmetry and can be performed at weak coupling, their realization in string theory will be more transparent. Note, however, that in general this will have drawbacks. Since the global symmetry acts as an isometry on the Higgs branch, a deformation involving the current multiplet can lead to instabilities on the Higgs branch if no other deformation is turned on, as happens in the $E_1$ case.

### 2.1 Weakly coupled phase after SUSY breaking

Consider a 5d $\mathcal{N} = 1$ gauge theory with a simple gauge algebra $\mathfrak{g}$ coupled to $n_H$ hypermultiplets in representation $R_i$ of $\mathfrak{g}$, with $i \in \{1, 2, \cdots, n_H\}$. The Lagrangian for this system is comprised of three terms: the supersymmetric YM Lagrangian $\mathcal{L}_{\text{SYM}}$, a CS term together with its SUSY completion $\mathcal{L}_{\text{SCS}}$, and the hypermultiplet Lagrangian $\mathcal{L}_{\text{matter}}$:

$$\mathcal{L} = \mathcal{L}_{\text{SYM}} + \mathcal{L}_{\text{SCS}} + \mathcal{L}_{\text{matter}}. \tag{2}$$

These terms are respectively given by [30, 31]:

$$\mathcal{L}_{\text{SYM}} = \frac{1}{g^2} \text{tr} \left( -\frac{1}{2} F_{\mu\nu} F^{\mu\nu} - \mathcal{D}_\mu \phi \mathcal{D}^\mu \phi + \frac{i}{2} \mathcal{D}_\mu \bar{\lambda} \gamma^\mu \lambda - \frac{i}{2} \bar{\lambda} \gamma^\mu \mathcal{D}_\mu \lambda + D^i D^i + i \bar{\lambda} [\phi, \lambda] \right),$$

$$\mathcal{L}_{\text{SCS}} = \frac{\kappa}{24\pi^2} \text{tr} \left[ AF^2 + \frac{i}{2} A^3 F - \frac{1}{10} A^5 - 3 \bar{\lambda} \gamma^{\mu\nu} \lambda F_{\mu\nu} + 6i \bar{\lambda} \sigma^i D^i \lambda \right] + \frac{g^2 \kappa}{2\pi^2} \text{tr} [\phi \mathcal{L}_{\text{SYM}}], \tag{3}$$

$$\mathcal{L}_{\text{matter}} = -\left| \mathcal{D}_\mu q \right|^2 + i \bar{\psi} \gamma^\mu \mathcal{D}_\mu \psi - \bar{q} \phi^2 q + q \sigma^i \bar{q} D^i + \sqrt{2} \bar{\psi} \lambda q - \sqrt{2} \bar{q} \bar{\lambda} \psi + i \bar{\psi} \phi \psi,$$

where $i = \{1, 2, 3\}$ is SU(2)$_R$ index, $\sigma^i$ are the Pauli matrices, and we work with the Lorentzian metric $\eta = \text{diag}(-1, +1, +1, +1, +1)$. In the following, for the sake of notation, we will suppress the flavor indices associated with the matter fields.

The gauge theory has an associated U(1)$_I$ instantonic symmetry whose current appears in eq. (1). Introducing a background vector multiplet for the U(1)$_I$

$$\mathcal{V}_I = (\phi_I, \lambda_I, A_I, D_I). \tag{4}$$

It can be shown that the bottom component $\phi_I$ of the background vector multiplet couples to the highest component of the instantonic current multiplet, which corresponds to $g^2 \mathcal{L}_{\text{SYM}}$.

Then the VEV of $\phi_I$ can be interpreted as the inverse square of the gauge coupling $1/g^2$. If we additionally turn on a VEV for the highest component $D_I$ of the background $U(1)_I$ vector multiplet

$$\langle D_I^i \rangle = d^i, \tag{5}$$

the Lagrangian changes by

$$\delta_I \mathcal{L} = d^i \text{tr}\left( \frac{i}{4} \bar{\lambda} \sigma^i \lambda + \phi D^i \right), \tag{6}$$

and supersymmetry is broken.

We can also consider turning on a background vector multiplet for the flavor symmetry acting on the hypermultiplets

$$\mathcal{V}_F = (\phi_F, \lambda_F, A_F, D_F). \tag{7}$$

Turning on a VEV for the bottom component $\phi_F$ of $\mathcal{V}_F$ gives a supersymmetric mass to the hypermultiplets. On the other hand, a VEV for the highest component

$$\langle D_F^i \rangle = \tilde{d}^i, \tag{8}$$

leads to a SUSY breaking mass for the scalars of the hypermultiplet

$$\delta_F \mathcal{L} = q \sigma^i \bar{q} \tilde{d}^i, \tag{9}$$

while no tree-level mass term is generated for the fermionic components of the hypermultiplets. Turning on all SUSY breaking deformations, the potential for the scalars takes the following form:[3]

$$V = \text{tr}\left( -\frac{1}{g^2} D^i D^i - (d^i \phi + q \sigma^i \bar{q}) D^i - q \sigma^i \bar{q} \tilde{d}^i + \bar{q} \phi^2 q \right). \tag{10}$$

After eliminating the auxiliary scalars $D^i$, we obtain

$$V = \text{tr}\left[ \frac{g^2}{4} (d^i \phi + q \sigma^i \bar{q})^2 - q \sigma^i \bar{q} \tilde{d}^i + \bar{q} \phi^2 q \right]. \tag{11}$$

Let us now study the vacua associated with this potential. Clearly, $\phi = q = 0$ is an extremum. Its stability can be determined by looking at the second derivatives of the potential

$$\left. \frac{\partial^2 V}{\partial q \partial \bar{q}} \right|_{q=\phi=0} = \sigma^i \tilde{d}^i, \qquad \left. \frac{\partial^2 V}{\partial \phi^2} \right|_{\phi=q=0} = \frac{g^2}{2} d^i d^i. \tag{12}$$

From the first equation, we see that in the presence of a D-term VEV $\tilde{d}^i$ for the flavor multiplet $\mathcal{V}_F$, there is a tachyonic instability, since the Pauli matrices have eigenvalues $\{+1, -1\}$. The negative mass of this mode can be compensated by turning on a VEV for $\phi_F$, namely a supersymmetric mass for the entire hypermultiplet. On the other hand, setting $\tilde{d}^i = 0$, the potential in eq. (12) is, at the origin of the Coulomb branch, completely unaware of the presence of the SUSY breaking parameter $d^i$. Consequently, a Higgs branch opens up exactly as in the supersymmetric case. Of course, having broken supersymmetry, the moduli space is expected to be lifted by quantum corrections. These effects are nevertheless expected to be suppressed in the regime we are dealing with, since the theory is IR-free and so the weakly coupled regime remains reliable.

In order to avoid this instability at weak coupling, in the following we will only consider (if not otherwise stated) D-term deformations involving only the instantonic symmetry of the theory. As we will see later on, these deformations will be naturally embedded in string theory as rotations of $(p, q)$ brane junctions outside the plane of the web.

---

[3]To simplify the analysis, we consider a vanishing CS level for the dynamical vector multiplet. We can always think of a non-vanishing CS level as the result of integrating out heavy fermions. This will be sufficient for our purposes.

## 2.2 Chern-Simons terms from UV dualities

An important tool to understand the dynamics of 5d supersymmetric theories comes from the prepotential. The moduli space of vacua of a 5d gauge theory consists of a Coulomb and a Higgs branch. The Coulomb branch is a real orbifold $\mathbb{R}^{\mathrm{rk}(\mathfrak{g})}/\mathcal{W}$, where $\mathrm{rk}(\mathfrak{g})$ denotes the rank of the gauge algebra, while $\mathcal{W}$ is the Weyl group of $\mathfrak{g}$. The low energy effective theory on the Coulomb branch is governed by the perturbative or Intriligator-Morrison-Seiberg (IMS) prepotential [3]

$$\mathcal{F} = \frac{1}{2} m_0 h_{ij} \phi^i \phi^j + \frac{\kappa}{6} d_{ijk} \phi^i \phi^j \phi^k + \frac{1}{12} \left( \sum_{r \in \Delta(\mathfrak{g})} |r \cdot \phi|^3 - \sum_{i=1}^{n_H} \sum_{w \in R_i} |w \cdot \phi - m_j|^3 \right), \quad (13)$$

where the first sum is over the set of roots $\Delta(\mathfrak{g})$ of the Lie algebra $\mathfrak{g}$, and the second sum is over the weights $w$ of the representation $R_i$ under which the $i$-th matter hypermultiplet transforms. $m_0$ is the inverse of the square of the YM coupling, $\kappa$ is the classical (bare) Chern-Simons level, and we have defined $h_{ij} = \mathrm{Tr}\left(T_i T_j\right)$, $d_{ijk} = \mathrm{Tr}\left(T_i \{T_j, T_k\}\right)$ in terms of the Cartan generators $T_i$ of the Lie algebra $\mathfrak{g}$.

In five dimensions, the vector multiplet can be dualized to a tensor multiplet containing a 2-form gauge field, whose electric sources are strings. Therefore, the Coulomb branch spectrum contains BPS monopole strings. The tensions of such strings are given by the first derivative of the prepotential with respect to the Coulomb branch moduli [1,3]

$$T_i = \frac{\partial \mathcal{F}}{\partial \phi^i}, \qquad i \in \{1, \cdots, \mathrm{rk}(\mathfrak{g})\}. \quad (14)$$

The low energy dynamics of type IIB $(p,q)$ webs, composed of junctions of $(p,q)$ 5-branes, are governed by 5d gauge theories. In this context, particles are described by generic $(p,q)$ strings connecting the various 5-branes composing the web. The masses of such particles are proportional to the length of the string junctions. Similarly, the monopole strings are realized as D3 branes which wrap a compact face of the 5-brane web. Their tension is proportional to the area of the face of the web that they wrap [2]. Hence, given a 5-brane web, there is a straightforward way to compute the prepotential associated with a given phase of the theory: one simply has to compute the areas of all the compact faces and solve the system of PDEs in eq. (14). In doing so, we will generate integration constants which, by dimensional analysis, must be cubic in the mass parameters $\{m_0, m_j\}$. These constants do not change either the metric on the CB nor the masses of the monopole strings, so they are often ignored in the literature. However, such integration constants carry important physical information on phase structures as discussed in [21]. Note that in the prepotential in eq. (13), Chern-Simons terms for the gauge fields appear as cubic terms in the CB parameters. Knowing the prepotential of the theory, it is sufficient to take three derivatives with respect to the CB parameters in order to extract the value of these CS coefficients. Similarly, we can interpret the mass parameters associated with the global symmetries of the theory as VEVs of the scalar field in the background vector multiplet, see the discussion following eq. (3). Then, we see that the presence of cubic terms in these mass parameters in the prepotential will signal the existence of CS terms for the background vector multiplets. Invariance under large gauge transformations imposes an integer quantisation condition on the CS coefficient. Thus, the CS level cannot be changed continuously and so is a rather robust quantity along the RG flow. Therefore, a change in the CS level is typically accompanied by a phase transition. In this way, knowing the global CS terms of the theory, one can detect the presence of phase transitions in the corresponding phase diagram.

Given that the constant terms do not affect the perturbative dynamics of the theory or the masses of the monopole strings, one may wonder if there is a way to uniquely fix them.

The answer turns out to be positive if we are dealing with different gauge theory descriptions descending from the same UV fixed point. In the 5d literature, theories enjoying this property are referred to as UV duals. They can be understood as different mass deformations of the same UV fixed point. When the mass deformation remains small (compared with the scale of the CB parameters), the prepotentials of the two theories should be the same since they describe the same UV SCFT on the CB. Note that, in this context, the prepotential will not necessarily be the perturbative prepotential of the two gauge theories: in going from the weakly coupled limit to the strong coupling limit, some non-perturbative statecan flip the sign of its mass,[4] contributing to the prepotential of the gauge theory and changing it. The corresponding phase is denoted in the literature as the CFT phase [32] and the corresponding prepotential as the CFT prepotential. In this phase, the two prepotentials coming from the two different gauge theory descriptions must agree.

The CB and mass parameters of the two theories will be mapped to each other in some non-trivial way. The UV dualities are often inherited from the S-duality of the type IIB string theory. As such one can extract the duality map by comparing the parametrisation of the $(p, q)$-web of a given theory with its S-dual. Requiring the prepotential to be invariant under UV dualities (namely under the transformations of the CB and mass parameters) is sufficient to partially (in certain cases completely) determine the constant terms in the prepotential.

A far more powerful object is the *complete* prepotential [32], which captures all the gauge theory phases, in addition to non-trivial information such as the global symmetry of the fixed point theory. In this context, the dynamics of the CFT phase of the theory is described in terms of CB parameters that are invariant under UV self-dualities, making the global symmetry of the CFT point manifest. These parameters are denoted in the literature as the invariant CB parameters and can be obtained systematically by knowing the CFT prepotential, as we will see later in detail. In addition, the complete prepotential is augmented by a hypermultiplet contribution, which encodes all the perturbative and non-perturbative hypermultiplets that can change their sign if we start from the CFT phase and mass deform away from the fixed point. Knowing the complete set of hypermultiplets is in general challenging, as we will later see. However, in the following, we will mainly be interested in the CFT part of the complete prepotential. Indeed, the global symmetry of the fixed point can in some cases fix completely the constant terms in the CFT prepotential and allow us to determine the CS levels of the global symmetry.

**On overall normalization of mass parameters**

Another subtle feature of the prepotential is the overall normalization of the mass parameters appearing in eq. (13). At present, several choices of normalization of the coefficient $m_0$ of the quadratic term in the prepotential of eq. (13) appear in the literature. In the absence of the integration constants mentioned above, the choice of overall normalization seems harmless. However, we emphasize that the integration constants appearing in the prepotential must be cubic in the masses $\{m_0, m_j\}$ and so represent CS terms for the global symmetries of the theory. As a consequence, the coefficients of these terms must be integers in order to have well-defined CS levels. We can then try to formulate a criterion in order to choose the correct normalization for the mass parameters. We expect this criterion to come from the spectrum of the theory. Looking at BPS operators charged under both the global and the gauge symmetry of the theory, we know that their mass $M$ is determined in terms of their charges $q_e^i, q_j$ by the BPS mass formula

$$M = |q_e^i \phi_i + q^j m_j|. \tag{15}$$

---

[4]Although, in all the cases we will consider, the CFT phase will coincide with the perturbative phase of the theory.

Changing the normalization of the masses will then change the normalization of the charges $q^j$. Integrating out such a state will lead to CS terms for both the gauge and the global symmetries. However, if the charges are not integer quantized, the CS terms will acquire non-integer levels. So, the minimal physical requirement that we should ask is to have correctly quantized charges. This is a necessary, but not sufficient, condition to have integer levels in the various phases of the theory. However, as we will see later on, in some cases this physical requirement will also fix the CS levels of the CFT prepotential to be integers.

To have correctly normalized charges, we need to take care of two additional phenomena that we can encounter: the mixing on the CB between the electric and the global symmetry charges and the enhancement of the global symmetry at the fixed point. In order to clearly explain these phenomena, we will consider two simple examples, namely the mixing in the $E_1$ theory and the global symmetry enhancement of the $E_2$ theory.

**Mixing phenomena**

The mixing phenomenon of the $E_1$ theory was first analyzed in [1]. The charge associated with the instantonic symmetry was shown to be a linear combination of the electric charge and the charge of the Cartan generator associated with the $SO(3)_I$ symmetry of the fixed point. In the language of the CFT prepotential, this mismatch comes from the difference between the invariant CB parameter and the CB parameter associated with the low energy description (in this case for positive $h$). The invariant CB parameter is[5]

$$\Phi := \phi + \frac{h^{E_1}}{2}, \tag{16}$$

where $\phi$ is the CB parameter of the gauge theory description. A point of the CB of the CFT is described by a VEV of the invariant CB parameter, while the CB associated with SU(2) SYM theory is described simply by $\phi$. Masses of BPS states have different expressions depending on which variables we use, either the CFT or the weakly coupled ones. In the first case, a generic BPS mass reads

$$M = |\tilde{q}_e \Phi + q_c h^{E_1}|, \tag{17}$$

where $q_c$ represents the charge associated with the Cartan subgroup of the $SO(3)_I$ global symmetry. On the other hand, at weak coupling, masses are measured in terms of the SU(2) SYM mass parameters

$$M = |q_e \phi + q_I h^{E_1}|. \tag{18}$$

Equating the two descriptions, we see that

$$q_e = \tilde{q}_e, \quad q_I = q_c + \frac{1}{2}\tilde{q}_e, \tag{19}$$

leading to a mixing between the charge associated with the Cartan subgroup of the $SO(3)_I$ global symmetry and the electric charge. So, the mixing phenomenon is present whenever the invariant CB parameter does not coincide with the CB parameter. In the following, we will insist on requiring the charges under all abelian symmetries to be integer quantised, both in the weak coupling and CFT variables. This translates to a criterion for determining the overall normalisation of mass parameters appearing in the prepotential. To see this recipe in action,

---

[5]The duality map for this case is $\phi \to \phi + h^{E_1}$, $h^{E_1} \to -h^{E_1}$.

consider the 5-brane web for pure SU(2) SYM theory

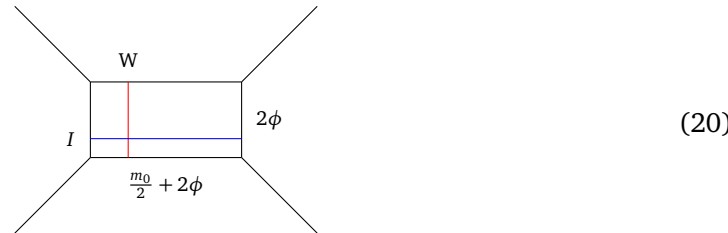 (20)

The above parametrisation leads to the following prepotential for SU(2) SYM

$$\mathcal{F} = \frac{m_0}{2}\phi^2 + \frac{4}{3}\phi^3 \,, \tag{21}$$

while the mass of the W boson and instanton particle are given by the lengths of the F1 (resp. D1) strings indicated in the web of figure (20)

$$m_W = |2\phi|\,, \quad m_I = \left|\frac{m_0}{2} + 2\phi\right| \,. \tag{22}$$

We can also express them in terms of the invariant Coulomb modulus

$$m_W = \left|2\Phi - \frac{m_0}{4}\right|\,, \qquad m_I = \left|2\Phi + \frac{m_0}{4}\right| \,. \tag{23}$$

Following the criterion mentioned above, we seek to rescale $m_0$ in such a way that the BPS states carry integer charges. The minimum choice that satisfies this criterion is to set $m_0 = 4h^{E_1}$. This is precisely the choice of normalization of [21] which leads to the correct normalization for the $U(1)_I^3$ CS levels.

**Global symmetry enhancement**

Another phenomenon is the enhancement of global symmetry at the fixed point for symmetry groups of rank higher than one. The mass parameters associated with the enhanced symmetry $G_{UV}$ at the fixed point can differ in general from the mass parameters associated with the weak coupling symmetry $G_{IR}$. The former will be linear combinations of the latter as prescribed by the embedding $G_{IR} \subset G_{UV}$. The simplest example of this phenomenon comes from the $E_2$ theory, the UV completion of the SU(2) + 1**F** theory. At weak coupling, the theory possesses a $U(1)_I \times SO(2)_F$ global symmetry with associated mass parameters $m_0, m_1$ respectively. At the fixed point, however, the global symmetry enhances to $SU(2)_I \times U(1)$ [33] and the associated mass parameters $x, y$ are actually linear combinations of the weak coupling parameters [32]

$$x = \frac{1}{4}(m_0 + m_1)\,, \quad y = -\frac{1}{4}(m_0 - 7m_1) \,. \tag{24}$$

On top of the enhancement of the global symmetry, also a mixing phenomenon can appear. In this case, mass parameters need to be normalized in order to have correctly quantized charges whenever we measure the masses of the BPS states in the $G_{UV}$ or the $G_{IR}$ parameters and in temrs of the weakly coupled or the strongly coupled CB parameters.

## 3 Super Yang-Mills theories

In this section, we consider SUSY-breaking deformations of UV completions of pure SYM theories. We consider only theories that have a known UV dual description, which is a key ingredient in detecting a phase transition. In particular, we first focus on the $SU(N)_N$ pure gauge

theories, starting with the $N = 3$ case and then extending the discussion to generic $N$. In both cases, we determine the CS levels of the global symmetry and the complete prepotential. These theories enjoy many common features with the $E_1$ theory. Firstly, at strong coupling, their symmetry enhances $SO(3)_I$ [33]. Secondly, there are no additional hypermultiplets that we can flop when turning on the mass deformation for the fixed point. This will facilitate our calculation of the complete prepotential of these theories. At infinite coupling the Higgs branch is $\mathbb{C}^2/\mathbb{Z}_2$ [34], exactly as in the $E_1$ case. Since the instability given by the non-SUSY deformation was driven by the Higgs branch, we see that these theories are also plagued by the same instability, leading to a phase diagram analogous to the $N = 2$ case.

We also study the $SU(3)_7$ pure gauge theory, which was shown to be UV dual to $G_2$ [35]. Since this theory does not enjoy self-duality, the CS levels are partially fixed by the duality. Nevertheless, we show the existence of a jump of the CS levels between the low energy $G_2$ phase and the $SU(3)_7$ YM phase, resulting from mass deforming the corresponding SYM theories.

## 3.1 $SU(3)_3$

Let us consider 5d $\mathcal{N} = 1$ SU(3) YM-CS theory with level 3. The gauge theory has a $U(1)_I$ global symmetry which is the remnant of the $SO(3)_I$ symmetry of the UV fixed point theory. Moreover, the Higgs branch of the UV fixed point is $\mathbb{C}^2/\mathbb{Z}_2$. The symmetry acting on the Higgs branch is $SO(3)_I$, rather than $SU(2)_I$, since all chiral ring generators transform under projective representations of the $\mathfrak{su}(2)_I$ Lie algebra. There is also an $SU(2)_R$ symmetry required for the consistency of the superconformal algebra. Finally, the theory enjoys a $\mathbb{Z}_3$ 1-form symmetry, which acts on Wilson lines in the gauge theory limit, and is expected to persist all the way to the UV fixed point as argued in [36].

The theory in question admits a brane web construction, that at a generic point of the Coulomb branch phase is given by

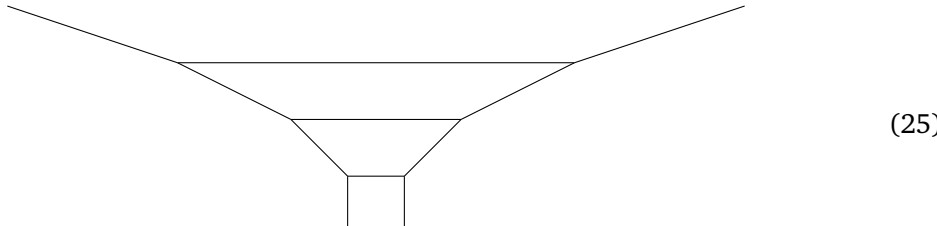

(25)

Note, in particular, that the symmetries of the theory are manifest in the web. The $SO(3)_I$ is realized as the symmetry on the two $[0, 1]$ 7-branes that can be attached to the NS5 branes, while the $SU(2)_R$ symmetry is identified with the isometry group of rotations in the three directions transverse to the 5-brane web. Finally, the $\mathbb{Z}_3$ 1-form symmetry is detectable by noticing that three fundamental strings terminating on the color (1,0) 5-branes can be screened by the external $(3, \pm 1)$ 5-branes, forming a string junction [37]. This is the analog of the screening of fundamental Wilson loops in the gauge theory language.

In order to achieve a UV dual description, we find it convenient to bring the web in figure (25) to a slightly different presentation by a combination of Hanany-Witten transition and

$T$-transformation of SL(2, $\mathbb{Z}$)

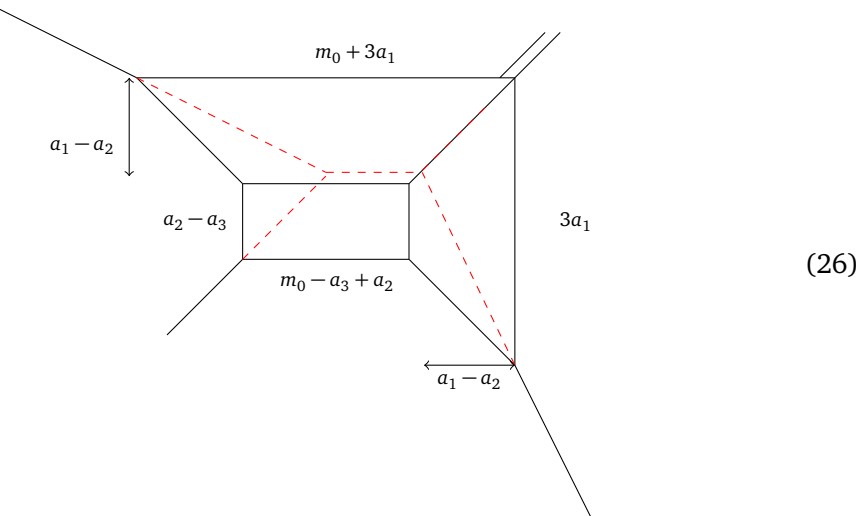

$$(26)$$

In this parametrization, the string tensions read

$$T_1 = (a_1 - a_2)(3a_1 + a_2 - a_3 + m_0), \qquad T_2 = (a_2 - a_3)(a_2 - a_3 + m_0). \tag{27}$$

Integrating the tensions, we obtain the IMS perturbative prepotential of SU(3)$_3$

$$\begin{aligned}
\mathcal{F}_{\text{SU}(3)_3} &= \frac{m_0}{2}(a_1^2 + a_2^2 + a_3^2) + \frac{1}{6}\left((a_1-a_2)^3 + (a_1-a_3)^3 + (a_2-a_3)^3\right) + \frac{1}{2}(a_1^3 + a_2^3 + a_3^3) \\
&= m_0(\phi_1^2 - \phi_1\phi_2 + \phi_2^2) + \frac{4}{3}\phi_1^3 + \phi_1^2\phi_2 - 2\phi_1\phi_2^2 + \frac{4}{3}\phi_2^3,
\end{aligned} \tag{28}$$

as can be shown by comparing it to the general expression in eq. (13) in the Weyl chamber $a_1 \geq a_2 \geq 0 \geq a_3$. In the second line of eq. (28), we change the parametrization of the CB going from the orthogonal to the Dynkin basis via the relations

$$a_1 = \phi_1, \qquad a_2 = -\phi_1 + \phi_2, \qquad a_3 = -\phi_2. \tag{29}$$

Let us now consider the S-dual configuration to the web in figure (26), shown below in figure (30). The low energy theory is still the SU(3)$_3$ theory, with parameters $\hat{a}_i$, $i = \{1, 2\}$ and $\hat{m}_0$.

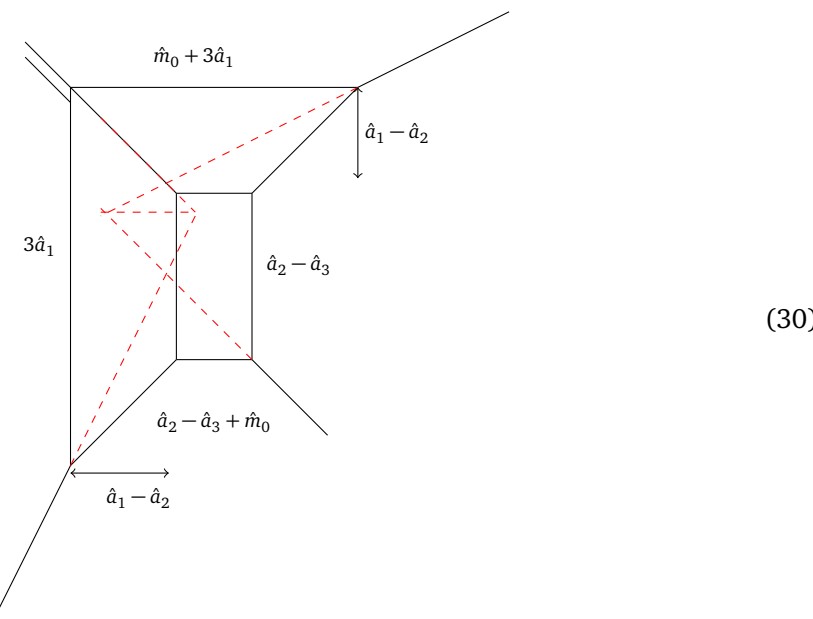

$$(30)$$

In this case, the tensions that we obtain from the web are given by

$$T_1 = (\hat{a}_1 - \hat{a}_2)(\hat{m}_0 + 2\hat{a}_1 - 2\hat{a}_3), \quad T_2 = (\hat{a}_2 - \hat{a}_3)(\hat{a}_2 - \hat{a}_3 + \hat{m}_0), \tag{31}$$

where in the above we have used the traceless condition $\sum_i \hat{a}_i = 0$, for the SU(3) CB parameters. Integrating the tensions, we can obtain, once again, the $SU(3)_3$ prepotential written in the new variables $\hat{a}_i$ and $\hat{m}_0$. The corresponding parameter map between the two UV dual descriptions is readily obtained by comparing the parametrizations in figures (25) and (30)

$$\hat{\phi}_1 = \phi_1 + \frac{1}{3}m_0, \qquad \hat{\phi}_2 = \phi_2 + \frac{2}{3}m_0, \qquad \hat{m}_0 = -m_0. \tag{32}$$

In terms of the global symmetry of the fixed point, we see that the Weyl group of $\mathfrak{su}(2)_I$ acts on the CB parameters by sending $\phi_i \to \hat{\phi}_i$, $i = \{1, 2\}$ and on the mass parameters as $m_0 \to \hat{m}_0 = -m_0$, as expected from the fact that $m_0$ represents the VEV of the lowest component of the background vector multiplet of the global symmetry group. Note, however, that the two perturbative prepotentials are not invariant under this reparameterization, owing to the fact that we have ignored the integration constant which arises in integrating the tensions in eq. (31).

In order to write down the prepotential invariant under the UV duality eq. (32), we can restore this integration constant $\alpha$ into the IMS prepotential

$$\mathcal{F} = m_0(\phi_1^2 - \phi_1\phi_2 + \phi_2^2) + \frac{4}{3}\phi_1^3 + \phi_1^2\phi_2 - 2\phi_1\phi_2^2 + \frac{4}{3}\phi_2^3 + \alpha m_0^3, \tag{33}$$

and require $\mathcal{F}$ to remain invariant under the duality map in eq. (32). The invariant prepotential is then given by

$$\mathcal{F} = m_0(\phi_1^2 - \phi_1\phi_2 + \phi_2^2) + \frac{4}{3}\phi_1^3 + \phi_1^2\phi_2 - 2\phi_1\phi_2^2 + \frac{4}{3}\phi_2^3 - \frac{1}{18}m_0^3. \tag{34}$$

In order for the CS level for the background vector multiplet of $U(1)_I$ to have the right quantisation, we rescale $m_0$ as

$$m_0 = 3h, \tag{35}$$

which yields

$$\mathcal{F} = 3h(\phi_1^2 - \phi_1\phi_2 + \phi_2^2) + \frac{4}{3}\phi_1^3 + \phi_1^2\phi_2 - 2\phi_1\phi_2^2 + \frac{4}{3}\phi_2^3 - \frac{9}{6}h^3, \tag{36}$$

from which we can read off

$$k_I = \frac{\partial^3 \mathcal{F}}{\partial h^3} = -9\,\mathrm{sgn}(h). \tag{37}$$

## 3.2 Generalisation to $SU(N)_N$

Generalizing the previous discussion to the $SU(N)_N$ case is fairly straightforward. Indeed, for generic $N$, we have the same global symmetry and Higgs branch at infinite coupling, and a similar embedding in string theory via a $(p, q)$ web construction.

**Complete prepotential**

The Weyl group of the global symmetry of the fixed point is related, in the $(p, q)$-web description, to S-duality. Looking at the parametrization of the web, we can easily see that the UV duality acts on the CB parameters as

$$\hat{a}_i = a_i + \frac{m_0}{N}, \quad \hat{m}_0 = -m_0 \qquad (i \in \{1, \cdots, N-1\}). \tag{38}$$

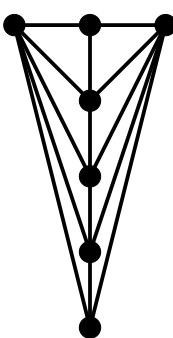

Figure 2: Grid diagram for the $SU(4)_4$ case.

Requiring the perturbative prepotential in the Weyl chamber $a_1 \geq a_2 \geq ... \geq a_{N-1} \geq 0 \geq a_N$ (resp. $\hat{a}_1 \geq \hat{a}_2 \geq ... \geq \hat{a}_{N-1} \geq 0 \geq \hat{a}_N$)

$$\mathcal{F} = \frac{m_0}{2}\sum_{i=1}^{N}a_i^2 + \frac{N}{6}\sum_{i=1}^{N}a_i^3 + \frac{1}{6}\sum_{i<j}^{N}(a_i - a_j)^3, \quad \sum_{i+1}a_i = 0, \tag{39}$$

to be invariant under UV duality fixes it completely to be of the form

$$\mathcal{F} = \frac{m_0}{2}\sum_{i=1}^{N}a_i^2 + \frac{N}{6}\sum_{i=1}^{N}a_i^3 + \frac{1}{6}\sum_{i<j}^{N}(a_i - a_j)^3 - \frac{(N-1)}{12N}m_0^3. \tag{40}$$

In order to have an integer CS level for the background $U(1)_I$, we perform the redefinition $m_0 = Nh$ and obtain

$$\mathcal{F} = \frac{Nh}{2}\sum_{i=1}^{N}a_i^2 + \frac{N}{6}\sum_{i=1}^{N}a_i^3 + \frac{1}{6}\sum_{i<j}^{N}(a_i - a_j)^3 - \frac{N^2(N-1)}{12}h^3, \tag{41}$$

from which we read the $U(1)_I^3$ CS level

$$k_I = -\frac{N^2(N-1)}{2}\,\text{sgn}(h), \tag{42}$$

which is integer for any $N$.

It is also possible to obtain the complete prepotential of the theory by following the procedure in [32]. Firstly, we calculate the effective coupling in the CFT phase of the theory. As already emphasized, this phase contains the CFT point, which can be reached by sending all the masses to zero. As a consequence, the $\mathcal{F}_{CFT}$ should be invariant under the action of the Weyl group of the enhanced global symmetry, as well as the correspoding effective couplings. In our case, since the theory has no flops, as can be inferred by looking at its grid diagram in figure (2), the CFT phase coincides with the perturbative phase. So, what we can do is just calculate the effective couplings from the perturbative prepotential

$$\frac{\partial^2 \mathcal{F}}{\partial a_i \partial a_i} = (2N + 4 - 2i)a_i + 2\sum_{k=1}^{i-1}a_i - 2a_N + 2m_0 \qquad (i = \{1, ..., N-1\}),$$

$$\frac{\partial^2 \mathcal{F}}{\partial a_i \partial a_j} = 2a_j - 2a_N + m_0 \qquad (i < j).$$

It is then easy to see that the effective couplings can be written in terms of linear combinations of

$$\hat{a}_i = a_i + \frac{1}{2N}m_0 \qquad (i = \{1, ..., N-1\}). \tag{43}$$

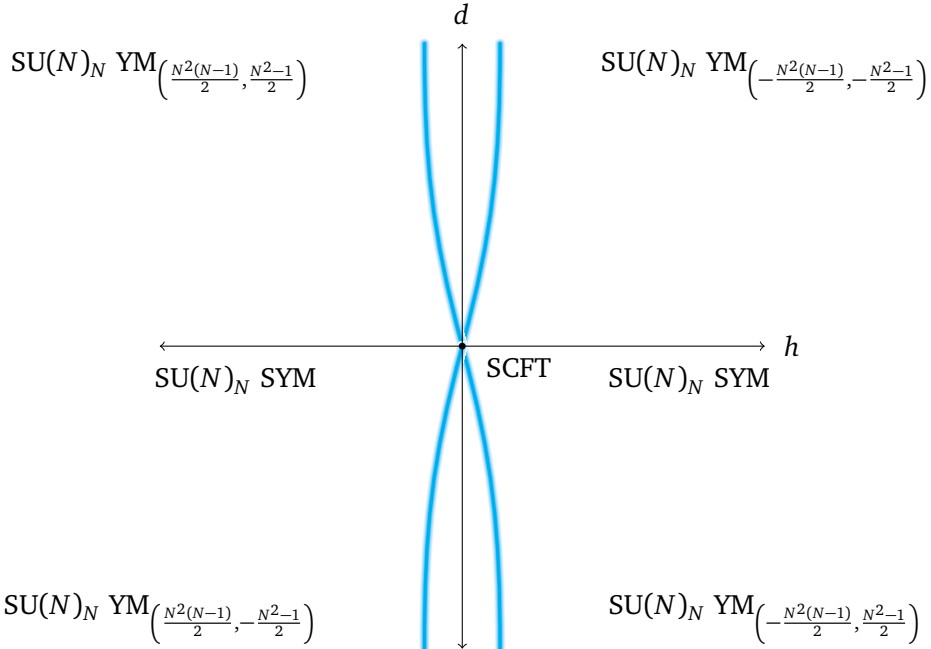

Figure 3: Summary of the phase diagram of the pure $SU(N)_N$ theory. The physics in each of the 4 quadrants is identical and related by the $\mathbb{Z}_2 \times \mathbb{Z}_2$ Weyl group action of the $SO(3)_I \times SU(2)_R$ symmetry. The fuzzy cyan line separates the symmetry broken phase from the pure YM ones. The exact shape of this line and the order of the phase transition taking place there is still not fully understood.

which can be shown to be manifestly invariant under the duality action induced by the Weyl group. The prepotential in the CFT phase can be obtained by integrating the effective couplings and matching the expression of the corresponding tensions with the perturbative ones, leading to the complete prepotential

$$\mathcal{F}_{CFT} = \frac{1}{6} \sum_{i<j} (\hat{a}_i - \hat{a}_j)^3 + \frac{N}{6} \sum_{i=1}^{N} \hat{a}_i^3 + \frac{1}{4} m_0 \hat{a}_N , \tag{44}$$

which is completely fixed by the symmetry of the fixed point. By expanding the invariant CB parameters, it is easy to obtain the value of the CS terms associated with the weakly coupled descriptions written in eq. (42).

**Phase diagram**

Exploiting both calculations and the weak coupling analysis of section 2, we obtain the generalisation of the phase diagram of the $E_1$ theory depicted in figure (3). In particular, we see that the CS term for the $U(1)_R$ symmetry depends on $N$. This comes from the fact that the D-term breaking deformation gives a mass to the gauginos. These are in the fundamental of $SU(2)_R$. When this symmetry is broken to its Cartan subgroup by the deformation, we obtain $N^2 - 1$ BPS particles charged under the $U(1)_R$ global symmetry, since the gauginos are in the adjoint[6] of $SU(N)$. Introducing a background gauge field for $U(1)_R$, it is easy to see that when the gauginos are integrated out, they generate the following CS level for the R-symmetry

$$k_R = -\frac{N^2 - 1}{2} \operatorname{sgn}(d) . \tag{45}$$

---

[6]For this reason, there is no shift for the CS level of the dynamical gauge field, since $d_{abc} = 0$, for the adjoint representation.

As in the $E_1$ case, the Weyl groups of $SU(2)_R$ and $SO(3)_I$ relate the CS levels for the $U(1)_I$ and $U(1)_R$ symmetries in the various quadrants of the $(h, d)$ plane, as shown in figure (3). The total jump of the CS levels across the $d$ axis is then

$$\Delta CS_I = -N^2(N-1), \qquad \Delta CS_R = -(N^2-1). \tag{46}$$

As stated in the $E_1$ case, we expect an instability on the HB of the SCFT deformed by the SUSY breaking deformation, since the structure of the Higgs branch and of the deformation is identical to the $E_1$ case. This leads to a symmetry breaking phase surrounding the $d$ axis, which is separated from the symmetry preserving phase by a phase transition at finite coupling, where the tachyon instability on the HB can be in principle resolved as shown in [23]. The phase diagram is then a natural generalisation of the $E_1$ phase diagram.

The previous picture is consistent with the string theory realization of this same deformation. This is completely analogous to the $E_1$ web in the parallel presentation [23]. The tachyon instability on the HB translates into an instability of the strings connecting the 7-branes hosting the global symmetry. This gets resolved when the $(p, q)$ web is opened by turning on the gauge coupling.

**Consistency with 't Hooft anomaly matching**

The $SU(N)_N$ theory is known to have a mixed 't Hooft anomaly between the zero-form instanton symmetry $U(1)_I$ and one-form symmetry $\mathbb{Z}_N^{(1)}$ [22]. Indeed, if we turn on the $U(1)_I$ background gauge field $A$ and $\mathbb{Z}_N^{(1)}$ background gauge field $B$, then the partition function of the theory on $S^1 \times \Sigma_4$ changes under the large gauge transformation of $U(1)_I$ with a unit winding as

$$Z_{SU(N)_N}[A, B] \ \rightarrow \ Z_{SU(N)_N}[A, B] \exp\left(-\frac{2\pi i}{2N}\int_{\Sigma_4} \mathcal{P}(B)\right), \tag{47}$$

where $\mathcal{P}(B)$ is the Pontryagin square.[7] The 't Hooft anomaly can be cancelled by coupling the theory to a bulk topological theory with the partition function[8]

$$\exp\left(\frac{2\pi i}{2N}\int_{Y_6} \frac{dA}{2\pi} \mathcal{P}(B)\right), \tag{48}$$

with $\partial Y_6 = M_5$. The 't Hooft anomaly is invariant along RG flow preserving $U(1)_I$ and $\mathbb{Z}_N^{(1)}$, and implies that the phase cannot be trivial. This is consistent with our proposal on the phase diagram in figure (3).

### 3.3 Duality between $G_2$ and $SU(3)_7$

Let us now consider the duality between the pure $G_2$ SYM and $SU(3)_7$ SYM. This was studied in great detail in [32, 42]. The common UV fixed point enjoys only the $U(1)_I$ instantonic symmetry preserved in the gauge theory phase. This can be seen from the isometry acting on the Higgs branch chiral ring, which is $U(1)$, since the HB is[9] $\mathbb{C}^2/\mathbb{Z}_4$. The IMS prepotential for a pure $G_2$ gauge theory, in the Dynkin basis, is given by [35]

$$\mathcal{F} = m_0(\phi_1^2 - 3\phi_1\phi_2 + 3\phi_2^2) + \frac{4}{3}\phi_1^3 - 4\phi_1^2\phi_2 + 3\phi_1\phi_2^2 + \frac{4}{3}\phi_2^3 + \frac{\alpha}{6}m_0^3, \tag{49}$$

---

[7]Throughout discussion on 't Hooft anomaly in this paper, we take $\Sigma_4$ as a spin manifold on which the integral of the Pontryagin square is even integer. This point will be more relevant in other cases.

[8]There are some possibilities on interpretations of the 't Hooft anomaly at the superconformal point [22, 33] involving two-group symmetries [33, 38–41].

[9]In general the Higgs branch of 5d $\mathcal{N} = 1$ SYM theory with gauge algebra $\mathfrak{g}$ is the orbifold $\mathbb{C}^2/\mathbb{Z}_{\hat{h}}$, where $\hat{h}$ is the dual Coxeter number of $\mathfrak{g}$ [43].

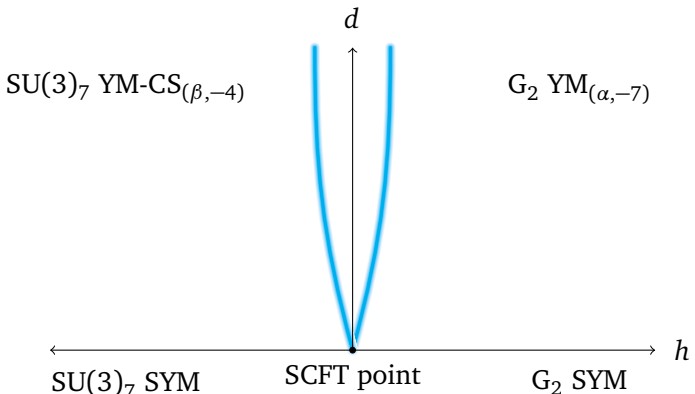

Figure 4: Summary of the phase diagram for the $G_2 - SU(3)_7$ case without matter. The negative $d$ region is omitted since it is related to the upper-half plane by the $\mathbb{Z}_2$ residual Weyl symmetry of $SU(2)_R$.

where we introduced an integration constant, which was neglected in [35]. The prepotential for pure $SU(3)_7$ theory, obtained from the S-dual web diagram in [44], in the Dynkin basis, is

$$\mathcal{F} = \hat{m}_0(\hat{\phi}_1^2 - \hat{\phi}_1\hat{\phi}_2 + \hat{\phi}_2^2) + \frac{4}{3}\hat{\phi}_1^3 + 3\hat{\phi}_1^2\hat{\phi}_2 - 4\hat{\phi}_1\hat{\phi}_2^2 + \frac{4}{3}\hat{\phi}_2^3 + \frac{\beta}{6}\hat{m}_0^3, \tag{50}$$

where we introduced the analogous integration constant $\frac{\beta}{6}m_0^3$ which was absent in [44]. The duality map between these theories, was worked out by comparing the parameterizations of the same web diagram and reads:

$$\hat{m}_0 = -\frac{m_0}{3}, \quad \hat{\phi}_1 = \phi_2 + \frac{m_0}{3}, \quad \hat{\phi}_2 = \phi_1 + \frac{2}{3}m_0. \tag{51}$$

In both cases, the CFT prepotential still coincides with the perturbative phase. So, requiring the prepotential for $SU(3)_7$ in eq. (28) to coincide with that of pure $G_2$ in eq. (49) upon substituting in the duality map in eq. (51) imposes a constraint on the integration constants

$$\beta = -27\alpha - 6. \tag{52}$$

This shows that, although the duality does not constrain completely the prepotential, it constrains the integration constants of the $SU(3)_7$ prepotential in terms of the $G_2$ one. The jump of the $U(1)_I$ CS level reads then

$$\Delta CS = -28\alpha - 6. \tag{53}$$

We see that, for any integer $\alpha$, there is a jump in the level between the two weakly coupled phases.

Also, the R-symmetry level jumps between the two weakly coupled phases. Integrating out the gaugino induces a shift of the CS level for the background gauge field $A_R$ of the $U(1)_R$ symmetry, which in the $G_2$ side reads

$$k_R = -7\,\text{sgn}(d), \tag{54}$$

while in the $SU(3)_7$ theory is

$$\tilde{k}_R = -4\,\text{sgn}(d). \tag{55}$$

Altogether, our analysis suggests the minimal phase diagram in figure (4) with a total jump of the CS levels across the $d$ axis of

$$\Delta CS_I = -28\alpha - 6, \qquad \Delta CS_R = -3. \tag{56}$$

The qualitative difference from the $SU(N)_N$ case is the absence of the reflection symmetry around the $d$-axis. This is expected, since the theory is not self-dual, so the SUSY deformations driving the SCFT point to the IR free gauge theory descriptions are not related by the action of the global symmetry. This is also the reason why the CS level cannot be completely fixed.

## 3.4 SU-Sp duality

In this section, we consider another higher rank generalisation by focusing on the UV duality between $SU(N+1)_{N+3}$ and $Sp(N)_{(N+1)\pi}$ SYM. As in the $G_2 - SU(3)_7$ case, we don't expect to be able to fix completely the prepotential of the theory, since there is no self-duality that we can employ. However, the duality will still be sufficient to detect the jump of the CS level as we go from one weakly coupled description to another. In particular, these theories do not show symmetry enhancement at the fixed point and the global 0-form symmetry remains just the instantonic $U(1)_I$ enjoyed by the weak coupling description. In addition, when $N$ is odd, there is $\mathbb{Z}_2$ 1-form symmetry [38], which coincides with the centre of $Sp(N)$, or equivalently (in the dual frame) the $\mathbb{Z}_{\gcd(N+1,N+3)} = \mathbb{Z}_2$ subgroup of the $\mathbb{Z}_{N+1}$ centre of $SU(N+1)$. There is no 1-form symmetry when $N$ is even. The Higgs branch of the SCFT also depends on the parity of $N$; for even $N$ there is no continuous Higgs branch,[10] while for odd $N$ the Higgs branch is $\mathbb{C}^2/\mathbb{Z}_{N+1}$. Consequently, we expect the instability on the Higgs branch to lead to a symmetry breaking phase for the case when $N$ is odd. On the other hand, the same instability cannot appear when $N$ is even, which makes these theories a good candidate to host a CFT at infinite coupling.[11]

As we will see, these theories are not expected to have any associated hypermultiplets contributing to the prepotential. For this reason, the computation of the complete prepotential is rather trivial: it is simply the perturbative one with the standard CB parameters. Indeed, we do not expect the prepotential to remain invariant under the UV duality transformation, but rather to be able to transform the $SU(N+1)_{N+3}$ prepotential into $Sp(N)_{(N+1)\pi}$ and vice versa.

In the following, in order to be pedagogical, we will first discuss the rank-2 case in some detail, before present the general results for arbitrary rank.

### 3.4.1 Duality between $SU(3)_5$ and $Sp(2)_\pi$

Let us start considering the rank-2 case, namely the $SU(3)_5$ theory. Its $(p,q)$ web is given by:

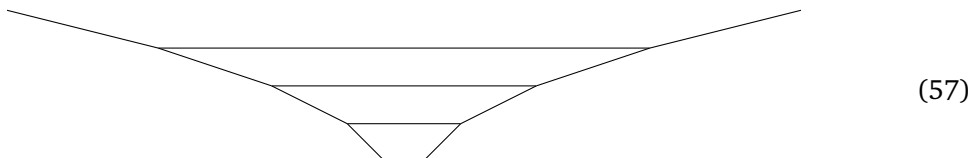

$$(57)$$

---

[10]More precisely, in this case the Higgs branch is a fat point. The only generator is the glueball superfield $S$ subject to the relation $S^2 = 1$ [43].

[11]This, however, does not exclude the existence of additional instabilities.

It is convenient to perform a couple of Hanany-Witten moves to bring the web into the following form

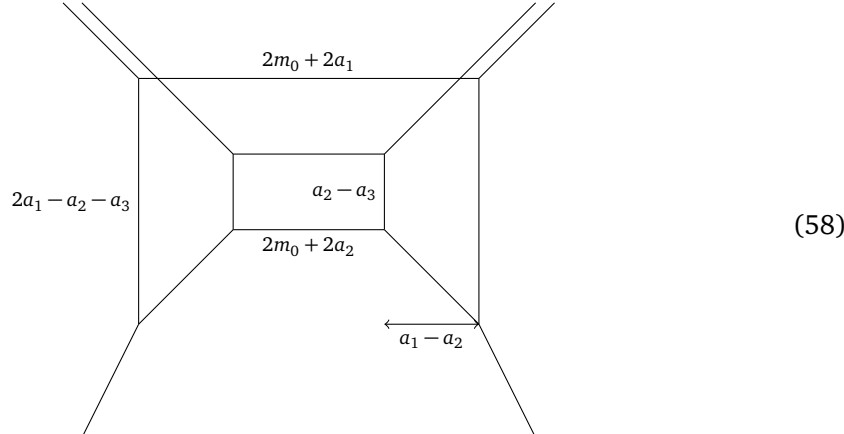

$$(58)$$

The corresponding tensions read

$$T_1 = \frac{\partial \mathcal{F}}{\partial \phi_1} = (a_1 - a_2)(2m_0 + 3a_1 + a_2 - 2a_3), \quad T_2 = \frac{\partial \mathcal{F}}{\partial \phi_2} = (2m_0 + 2a_2)(a_2 - a_3), \quad (59)$$

where as usual $a_i$, $i = \{1, 2\}$ represent the coordinate in the orthogonal basis, while $\phi_i$, $i = \{1, 2\}$ the ones in the Dynkin basis. Solving for the PDE in eq. (14), we obtain the prepotential for the SU(3)$_5$ theory

$$\mathcal{F}_{\text{SU}(3)_5} = 2m_0(\phi_1^2 - \phi_1\phi_2 + \phi_2^2) + \frac{1}{3}(4\phi_1^3 + 6\phi_1^2\phi_2 - 9\phi_1\phi_2^2 + 4\phi_2^3) + \frac{\alpha}{6}m_0^3, \quad (60)$$

where $\frac{\alpha}{6}m_0^3$ is an integration constant, to be fixed below. Let us now consider the S-dual web, given by

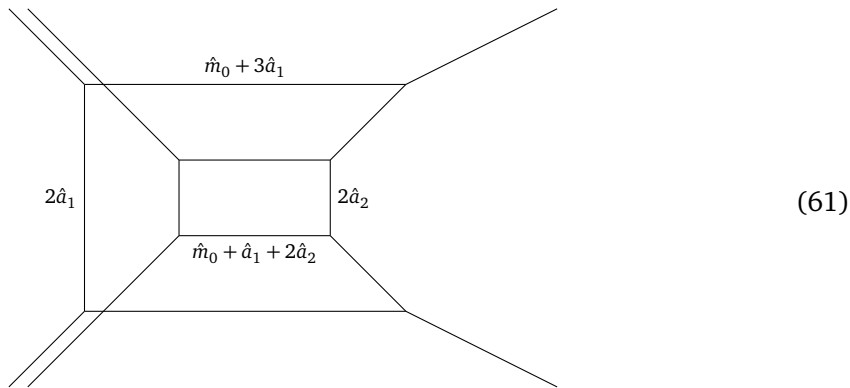

$$(61)$$

The tensions in this case read

$$T_1 = (\hat{a}_1 - \hat{a}_2)(2\hat{m}_0 + 5\hat{a}_1 + 3\hat{a}_2), \quad T_2 = 2\hat{a}_2(\hat{m}_0 + \hat{a}_1 + 2\hat{a}_2). \quad (62)$$

The solution to the PDE in eq. (14), with these tensions, leads to the prepotential for the Sp(2)$_\pi$ theory

$$\mathcal{F}_{\text{Sp}(2)_\pi} = \hat{m}_0(\hat{a}_1^2 + \hat{a}_2^2) + \frac{1}{6}(\hat{a}_1 - \hat{a}_2)^3 + \frac{1}{6}(\hat{a}_1 + \hat{a}_2)^3 + \frac{4}{3}(\hat{a}_1^3 + \hat{a}_2^3) + \frac{\beta}{6}\hat{m}_0^3. \quad (63)$$

Comparing the parameterizations of the webs in eqs. (58) and (61) leads to the following duality map

$$\hat{m}_0 = -3m_0, \quad \hat{a}_i = a_i + m_0 \qquad (i = \{1, 2\}). \quad (64)$$

The cubic $U(1)_I^3$ Chern-Simons term for the background vector multiplet is then partially fixed by requiring the prepotentials eqs. (60) and (63) to agree upon the above change of variables since the CFT point is contained in the perturbative phase. This imposes the following relation between the integration constants in eqs. (60) and (63)

$$\alpha + 27\beta + 12 = 0\,. \tag{65}$$

As anticipated above, UV duality is not enough to completely fix the CS terms. However, the jump of the level across the two phases may be fixed in terms of one of the CS levels and equals

$$\Delta\text{CS} = -28\beta - 12\,, \tag{66}$$

which is always different from zero for any integer $\beta$. Before discussing the phase diagram and the stability of the theory after the SUSY breaking deformation, it is worth generalizing this discussion to the rank-$N$ case.

### 3.4.2 Arbitrary rank generalisation

The brane web for $SU(N+1)_{N+3}$ is given by

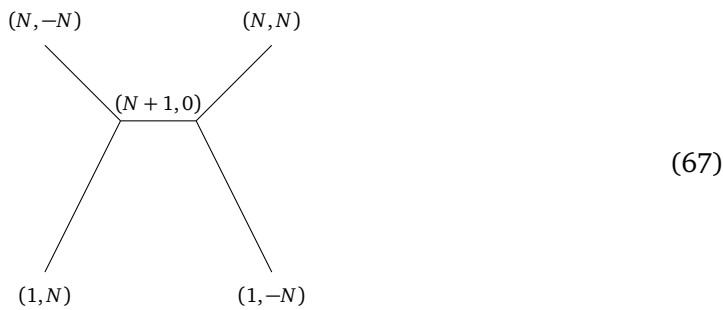

$$\tag{67}$$

It admits no flops, so the prepotential of the CFT phase coincides with the perturbative prepotential. Moreover, since there is no enhancement of the global symmetry, there is no notion of an invariant Coulomb branch parameter. The prepotential following from eq. (13) reads

$$\mathcal{F}_{SU(N+1)_{N+3}}(m_0, a_i) = m_0 \sum_{i=1}^{N+1} a_i^2 + \frac{N+3}{6} \sum_{i=1}^{N+1} a_i^3 + \frac{1}{6} \sum_{i<j}^{N+1} (a_i - a_j)^3 + \frac{\alpha}{6} m_0^3\,, \quad \sum_{i=1}^{N+1} a_i = 0\,, \tag{68}$$

where we have restored the integration constant $\frac{\alpha}{6} m_0^3$. The S-dual web corresponding to $Sp(N)_{(N-1)\pi}$ is given by

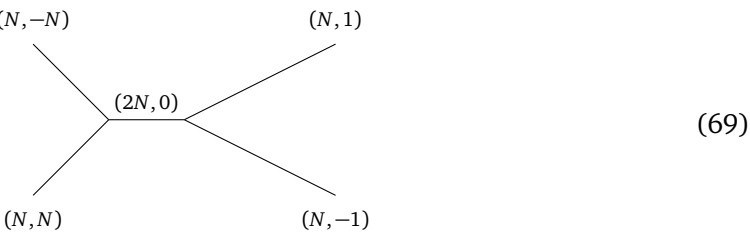

$$\tag{69}$$

from which one can extract the following prepotential

$$\mathcal{F}_{Sp(N)_{(N-1)\pi}}(\hat{m}_0, \hat{a}_i) = \hat{m}_0 \sum_{i=1}^{N} \hat{a}_i^2 + \frac{1}{6} \sum_{i<j}^{N} \left[ (\hat{a}_i - \hat{a}_j)^3 + (\hat{a}_i + \hat{a}_j)^3 \right] + \frac{4}{3} \sum_{i=1}^{N} \hat{a}_i^3 + \frac{\beta}{6} \hat{m}_0^3\,. \tag{70}$$

The duality map, in this case, reads

$$\hat{m}_0 = -(N+1)m_0\,, \qquad \hat{a}_i = a_i + m_0\,. \tag{71}$$

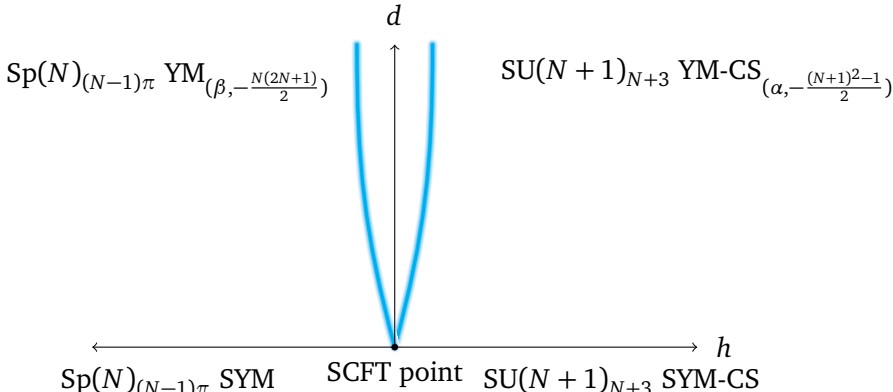

Figure 5: Summary of the phase diagram for the $SU(N+1)_{N+3}$-$Sp(N)_{(N+1)\pi}$ case without matters.

and can be inferred from the $(p,q)$ web associated with the two theories. Requiring the prepotential in eq. (70) to agree with eq. (68) under this duality map imposes a relation between the integration constants

$$\alpha + \beta(N+1)^3 + 2N(N+1) = 0.\qquad(72)$$

Again, as expected, the CS term is not completely fixed by the deformation. However, the jump is fixed in terms of $\beta$ as

$$\Delta CS = -\beta[(N+1)^3 + 1] - 2N(N+1).\qquad(73)$$

If we apply the deformation in eq. (6) to the two theories, the gauginos and the scalar gauginos get a mass and they can be integrated out. In the special unitary case, this gives a contribution to the R-symmetry CS level

$$k_R = -\frac{(N+1)^2 - 1}{2}\mathrm{sgn}(d),\qquad(74)$$

while in the symplectic case

$$\tilde{k}_R = -\frac{N(2N+1)}{2}\mathrm{sgn}(d).\qquad(75)$$

The corresponding phase diagram is summarised in figure 5 with a total jump of the CS levels across the $d$ axis of

$$\Delta CS_I = -\beta[(N+1)^3 + 1] - 2N(N+1),\qquad \Delta CS_R = \frac{N(N-1)}{2}.\qquad(76)$$

**Consistency with 't Hooft anomaly matching**

When $N$ is odd, the theories enjoy a $\mathbb{Z}_2^{(1)}$ one-form symmetry and it is interesting to ask whether there are mixed 't Hooft anomalies between $U(1)_I$ and $\mathbb{Z}_2^{(1)}$ as in the $SU(N)_N$ theory. This can be easily answered by using the results in [22, 38]. Again, turning on both background gauge fields and making the large gauge transformation of $U(1)_I$, the partition functions of the two theories are transformed as

$$Z_{SU(N+1)_{N+3}}[A,B] \rightarrow Z_{SU(N+1)_{N+3}}[A,B]\exp\left(\frac{2\pi i N(N+1)}{8}\int_{\Sigma_4}\mathcal{P}(B)\right),\qquad(77)$$

and

$$Z_{\mathrm{Sp}(N)_{(N-1)\pi}}[A,B] \;\to\; Z_{\mathrm{Sp}(N)_{(N-1)\pi}}[A,B]\exp\left(\frac{2\pi iN}{4}\int_{\Sigma_4}\mathcal{P}(B)\right). \tag{78}$$

There is an important difference between the $N = 4k + 1$ and $N = 4k + 3$ cases ($k \in \mathbb{N}$). For $N = 4k + 1$, both partition functions nontrivially transform with the same factor and therefore the 't Hooft anomaly matches on the both side. On the other hand, for $N = 4k + 3$, the transformation for the $\mathrm{SU}(N + 1)_{N+3}$ theory is trivial while the one for $\mathrm{Sp}(N)_{(N-1)\pi}$ is not.[12] One scenario to make this consistent with the duality is that the $\mathbb{Z}_2^{(1)}$ symmetries may be emergent and the 't Hooft anomalies do not have to match. Another possibility is that the gauge groups of the theories may have different global structures, namely the duality would be actually enjoyed by $\mathrm{SU}(N + 1)/\mathbb{Z}_2$ and $\mathrm{Sp}(N)/\mathbb{Z}_2$ without centers. In summary, if the $\mathbb{Z}_2^{(1)}$ symmetries are not emergent, then the 't Hooft anomaly for $N = 4k + 3$ implies that the phase cannot be trivial along RG flow and this is consistent with the phase diagram in figure 5.

# 4 The rank-$N$ $E_1$ SCFT

A natural generalisation of the $E_1$ theory is the so-called rank-$N$ $E_1$ SCFT, denoted as $E_1^{(N)}$. This is the UV fixed point of an $\mathrm{Sp}(N)$ gauge theory with a single anti-symmetric hypermultiplet. Contrary to the cases described above, this class of theories possesses a richer global symmetry and dynamics. Firstly, the global symmetry at the fixed point is $\mathrm{SO}(3)_I \times \mathrm{SO}(3)_A$, where the first group is associated with the enhancement of the instantonic $\mathrm{U}(1)_I$ symmetry of the $\mathrm{Sp}(N)$ theory and the second with the symmetry under which the anti-symmetric hypermultiplet transforms. Although two different mass parameters are present, as we will shortly see, no mixing will arise. In addition to the 0-form symmetry, we have also a $\mathbb{Z}_2$ 1-form symmetry, which in the weakly coupled description coincides with the center of the $\mathrm{Sp}(N)$ gauge algebra.

The presence of a global symmetry composed of two different groups leads to two possible mass deformations involving the global symmetry current multiplets. Due to stability reasons of the weakly coupled description, see section 2, in the following we only focus on the deformation involving the instantonic symmetry. Nevertheless, we briefly comment on the $(p,q)$ web constructions of the various possible non-SUSY deformations and their instability issues. The theory, due to the enhancement of the global symmetry, enjoys self-duality at the fixed point, so we expect the CS terms to be completely fixed by the symmetry of the theory. The Higgs branch of the theory at the fixed point is the moduli space of $\mathrm{SO}(3)$ $N$-instantons [45]. Away from the fixed point, the Higgs branch reduces to the $N$-th symmetric product of $\mathbb{C}_2 \times \mathbb{Z}_2$ arising from the anti-symmetric hypermultiplet and the glueball superfield $S$ subject to the relation $S^2 = 1$ [43]. In the following, we will briefly comment on the fate of this large moduli space after the non-SUSY deformations.

This theory is interesting independently of breaking supersymmetry thanks to a variety of reasons. First of all, the phase structure associated with its supersymmetric mass deformations is quite rich and was the subject of a recent study [46]. It would be interesting to reproduce their phase diagram by a direct study of the complete prepotential of the theory. When $N$ is large, this theory possesses a holographic dual, the Brandhuber-Oz solution [7]. Studying its non-SUSY deformation can be important, since predictions arising for the fate of this deformation in the large $N$ limit can be tested by turning on non-SUSY deformations in holography.

In the following, we will study the rank-2 case in great detail. The complete prepotential was already known in [32] from the decoupling of the $\mathrm{Sp}(2)+1\mathbf{AS}+7\mathbf{F}$. Here, we directly

---

[12]Note that the integral of the Pontryagin square is even for spin manifold.

compute it from the $(p,q)$ web and we generalize its CFT part to the rank-$N$ case. In this way, we calculate the CS terms for both the $U(1)_I$ and the $U(1)_A$ symmetries in the weakly coupled regime. Then, we mass deform via a non-SUSY deformation and study the jump of the background CS levels. Looking at the $(p,q)$ web construction of this deformation, we comment on the instability issues arising and we chart the phase diagram of this class of theories.

## 4.1 Rank-2 theory

Let us consider the 5d rank-2 $E_1$ SCFT. Activating a VEV for the bottom component of the background vector multiplet of the $SO(3)_I$ global symmetry takes us to the gauge theory phase, which is $Sp(2)_0+1\mathbf{AS}$, and breaks $SO(3)_I$ to the $U(1)_I$ topological symmetry of the field theory. The brane web for 5d $Sp(2)+1\mathbf{AS}$ is given by

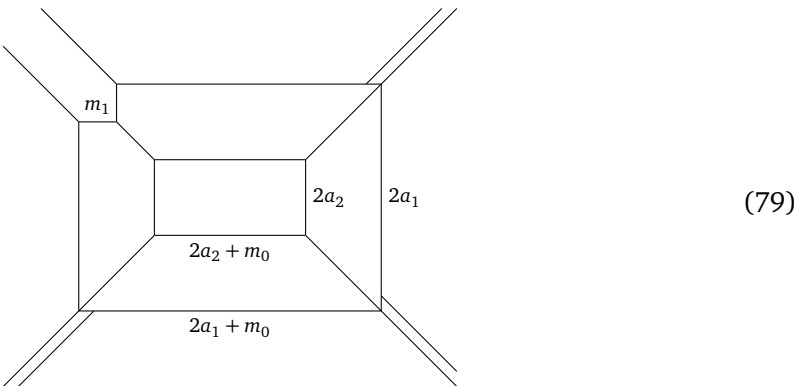

$$\tag{79}$$

From this diagram, we compute the string tensions to be

$$T_1 = 4(a_1^2 - a_2^2) + 2m_0(a_1 - a_2) - m^2, \qquad T_2 = 2a_2(m_0 + 2a_2). \tag{80}$$

Solving for the PDE in eq. (14) with the above tensions leads to the IMS prepotential

$$\mathcal{F}_{\text{IMS}} = \frac{4}{3}(a_1^3 + a_2^3) + m_0(a_1^2 + a_2^2) - m^2 a_1 + \alpha m_0^3 + \beta m_0 m^2, \tag{81}$$

where the CB independent terms represent our parameterization of the integration constants.[13]

Now consider the S-dual of the brane web in (79), namely the following diagram:

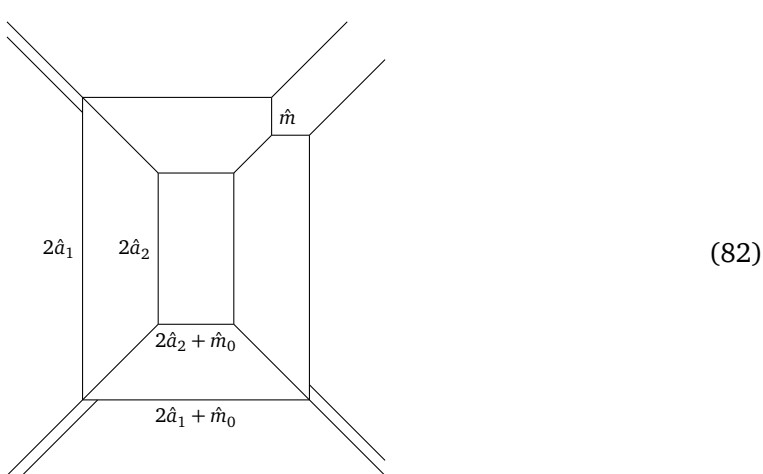

$$\tag{82}$$

---

[13]The reader may note that it is also perfectly reasonable to turn on additional background CS terms proportional to $m_0^2 m$ and $m^3$. However, both such terms are invariant under S-duality, and setting their coefficients to 0 is a perfectly consistent choice. Moreover, since these terms are invariant under S-duality, their levels will not jump as we go from the $h > 0$ phase to the $h < 0$ one.

Upon comparing the parameterization of the two web diagrams, we are led to the following duality map

$$\hat{a}_i = a_i + \frac{m_0}{2}, \quad \hat{m}_0 = -m_0, \quad \hat{m}_1 = m_1 \qquad (i = \{1,2\}). \tag{83}$$

Invariance of the prepotential under this duality map leads to the constraint

$$\alpha = -\frac{1}{12} - \frac{m_1^2}{4m_0^2} - \frac{m_1^2 \beta}{m_0^2}, \tag{84}$$

thus the duality invariant prepotential is

$$\mathcal{F} = \frac{4}{3}(a_1^3 + a_2^3) + m_0(a_1^2 + a_2^2) - m^2 a_1 - \frac{1}{12}m_0^3 - \frac{1}{4}m_0 m^2. \tag{85}$$

The background CS couplings are integers when we rescale

$$m_0 \to 2h, \tag{86}$$

which, as we will see, will be natural in order to correctly normalize the charges of BPS states in terms of the CB invariant parameters. The duality invariant prepotential thus reads

$$\mathcal{F} = 2h(a_1^2 + a_2^2) + \frac{4}{3}(a_1^3 + a_2^3) - a_1 m^2 - \frac{2}{3}h^3 - \frac{1}{2}hm^2. \tag{87}$$

In fact, we can go one step further, by directly calculating the complete prepotential of the theory. Starting from the perturbative prepotential in eq. (81), we need to introduce two invariant CB parameters, since the theory enjoys an $U(1)_I \times SO(3)_A$ global symmetry, that it is known to enhance to $SO(3)_I \times SO(3)_A$ at the fixed point. The CFT phase of the theory can be reached by demanding $a_1, a_2 \gg |m|, h$ and $a_1 - a_2 \gg |m|, h$, which are already satisfied by the prepotential in this phase. As it is clear from the lengths of the corresponding $(p,q)$ web in eq. (79), no additional hypermultiplet needs to be flopped in order to reach the CFT phase. Consequently, the effective couplings in the CFT phase read

$$\frac{\partial^2 \mathcal{F}}{\partial a_1^2} = 8\left(a_1 + \frac{1}{2}h\right), \quad \frac{\partial^2 \mathcal{F}}{\partial a_2^2} = 8\left(a_2 + \frac{1}{2}h\right), \tag{88}$$

and the invariant CB parameters are

$$\hat{a}_i = a_i + \frac{1}{2}h, \tag{89}$$

generalizing the $E_1$ case. From the tensions of the monopole strings, we can fix the linear terms in $\mathcal{F}$, obtaining the CFT prepotential

$$\mathcal{F}_{CFT} = -m^2 \hat{a}_1 + \frac{4}{3}\sum_{i=1}^{2} \hat{a}_i^3 - h^2 \sum_{i=1}^{2} \hat{a}_i. \tag{90}$$

The CS terms for the $U(1)_I \times U(1)_A$ groups are then

$$\frac{\partial^3 \mathcal{F}_{CFT}}{\partial h^3} = -4, \quad \frac{\partial^3 \mathcal{F}_{CFT}}{\partial h \partial m^2} = -1. \tag{91}$$

We see that the hypermultiplet masses in terms of the invariant CB parameters[14]

$$\frac{1}{6}[|\hat{a}_1 - \hat{a}_2 \pm m|]^3 + \frac{1}{6}[|\hat{a}_1 + \hat{a}_2 - h \pm m|]^3, \tag{92}$$

_______________

[14]Here we adopt the conventions of [32] defining

$$[|x|] := \theta(-x) \cdot x = \begin{cases} 0, & x > 0, \\ x, & x < 0. \end{cases}$$

have integer quantized charges, justifying the rescaling of $m_0$ we did above.

The hypermultiplet contribution to the complete prepotential can be obtained by acting on the masses of the anti-symmetric hypermultiplet with the $\mathbb{Z}_2$ Weyl group of $SO(3)_I$, which sends $h \to -h$. The following representations

$$\frac{1}{6}[|\hat{a}_1 - \hat{a}_2 \pm m|]^3 + \frac{1}{6}[|\hat{a}_1 + \hat{a}_2 \pm h \pm m|]^3 . \tag{93}$$

exhaust all possible hypermultiplet contributions to the theory. As it is clear from the previous procedure, the whole set of hypermultiplets are orbits of the $SO(3)_I$ Weyl group obtained from perturbative hypermultiplets. In this way, we can determine the complete prepotential, which reads

$$\mathcal{F}_{\text{compl.}} = -m^2 \hat{a}_1 + \frac{4}{3}\sum_{i=1}^{2} \hat{a}_i^3 - h^2 \sum_{i=1}^{2} \hat{a}_i + \frac{1}{6}[|\hat{a}_1 - \hat{a}_2 \pm m|]^3 + \frac{1}{6}[|\hat{a}_1 + \hat{a}_2 \pm h \pm m|] , \tag{94}$$

and matches the calculation in [32].

## 4.2 Generalisation to arbitrary rank

The discussion in the previous section can be (partially) generalized to arbitrary rank. The brane system is completely analogous and given explicitly in [47]. For instance, when $N = 3$ the web is given by

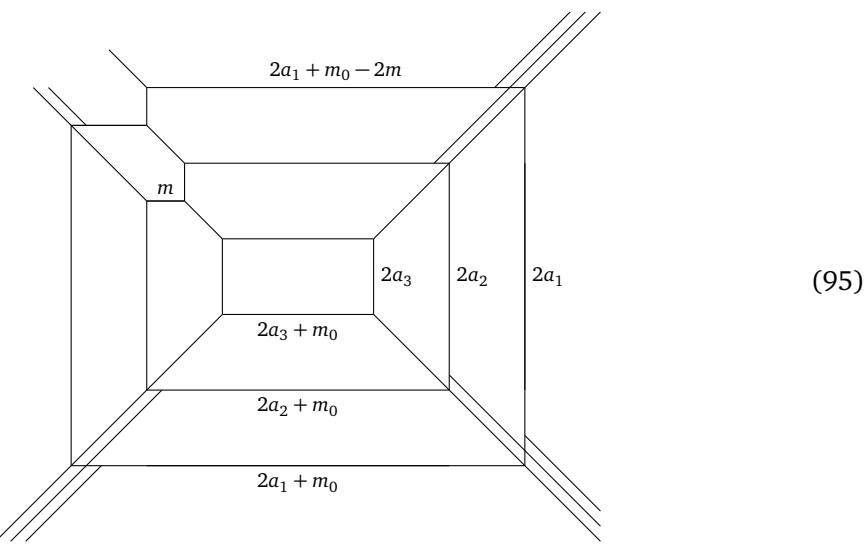

$$\tag{95}$$

The tensions read from the web diagram are

$$T_i = \frac{\partial \mathcal{F}_{\text{IMS}}}{\partial a_i} - \frac{\partial \mathcal{F}_{\text{IMS}}}{\partial a_{i+1}} = 2a_i(m_0 + 2a_i) - 2a_{i+1}(m_0 + 2a_{i+1}) - m^2 \qquad (i = \{1, ..., N-1\}),$$
$$\tag{96}$$

$$T_N = \frac{\partial \mathcal{F}_{\text{IMS}}}{\partial a_N} = 2a_N(m_0 + 2a_N).$$

Solving the PDE in eq. (14), with these tensions, we obtain the IMS prepotential

$$\mathcal{F}_{\text{IMS}} = m_0 \sum_{i=1}^{N} a_i^2 + \frac{4}{3}\sum_{i=1}^{N} a_i^3 - m^2 \sum_{i=1}^{N}(N-i)a_i + \alpha m_0^3 + \beta m_0 m^2 . \tag{97}$$

The UV duality induces the following map

$$\hat{a}_i = a_i + \frac{m_0}{2}, \qquad \hat{m}_0 = -m_0.$$
(98)

Requiring the prepotential to respect this duality imposes the following constraint on the integration constants

$$\alpha = -\frac{N}{24} - \frac{m^2}{m_0^2}\left(\beta + \frac{N(N-1)}{8}\right),$$
(99)

therefore the duality invariant prepotential for this theory is given by

$$\mathcal{F} = 2h\sum_{i=1}^{N} a_i^2 + \frac{4}{3}\sum_{i=1}^{N} a_i^3 - m^2\sum_{i=1}^{N}(N-i)a_i - \frac{2N}{6}h^3 - \frac{N(N-1)}{4}hm^2,$$
(100)

where we have performed the redefinition $m_0 = 2h$ to ensure integrality of the non-mixed CS level of $U(1)_I$.

On the contrary with respect to the $N = 2$ case, the construction of the complete prepotential is complicated by the presence of a large number of flops, testified by the intricated structure of the phase diagram unveiled in [46]. Nevertheless, since we are only interested in the perturbative phase, which is continuously connected to the CFT point without flops, we can just proceed to compute the CFT prepotential. Starting from the prepotential in eq. (81), the invariant CB parameters are easily obtained from the effective couplings as

$$\hat{a}_i = a_i + \frac{1}{4}m_0.$$
(101)

The tension of the monopole strings obtained from the corresponding $(p,q)$ web in figure (95) read

$$\frac{\partial \mathcal{F}}{\partial a_i} = 2a_i(m_0 + 2a_i) - (N-i)m^2, \quad \frac{\partial \mathcal{F}}{\partial a_N} = 2a_N(m_0 + 2a_N).$$
(102)

Matching the values of the tension, the CFT prepotential reads

$$6\mathcal{F}_{CFT} = -6m^2\sum_{i=1}^{N-1}(N-i)\hat{a}_i + 8\sum_{i=1}^{N}\hat{a}_i^3 - \frac{3}{2}m_0^2\sum_{i=1}^{N}\hat{a}_i,$$
(103)

generalizing naturally the $N = 2$ case. The CS terms then match the values obtained in eq. (100).

Looking at the masses of the perturbative hypermultiplets, written in terms of the invariant CB parameters

$$\frac{1}{6}\sum_{i<j}[|\hat{a}_i - \hat{a}_j \pm m|]^3 + \frac{1}{6}\sum_{i<j}[|\hat{a}_i + \hat{a}_j - h \pm m|]^3,$$
(104)

we see that the rescale we did above gives us the correctly quantized charges. Part of the hypermultiplet contribution to the complete prepotential can be obtained by applying the Weyl reflection to $m_0$

$$\frac{1}{6}\sum_{i<j}[|\hat{a}_i - \hat{a}_j \pm m|]^3 + \frac{1}{6}\sum_{i<j}[|\hat{a}_i + \hat{a}_j \pm h \pm m|]^3.$$
(105)

However, for generic $N$, the complexity of the phase diagram of the theory in terms of the mass parameters $(m_0, m)$ shown in [46] suggests the existence of additional representations constituted by purely non-perturbative states, which can be only identified as flops of the $(p,q)$ web.

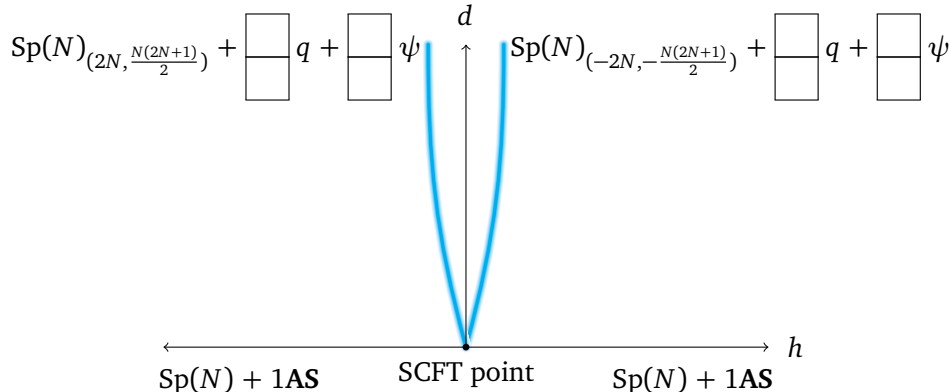

Figure 6: Phase diagram of the SUSY-broken $E_1^{(N)}$ theory. We omit drawing the lower half plane explicitly, since this may be recovered by applying the $\mathbb{Z}_2$ Weyl group of $SU(2)_R$ algebra.

We can now activate the SUSY breaking deformation directly in the weak coupling regime. Contrary to the previously studied cases, we have three choices: giving a VEV to the D-term for $SO(3)_A$, breaking SUSY and $SO(3)_A \to U(1)_A$, giving a VEV to the D-term of $U(1)_I$ or giving a VEV to both. However, as follows from the analysis in section 2, only the first option is stable at tree level, while the other two choices need to be supplemented by adding a supersymmetric mass, in order to avoid tachyonic instabilities. In the following, we will deform the theory only by turning on a deformation for $U(1)_I$.

The effects of this SUSY breaking deformation on the weakly coupled theory is the usual one: both the gauginos and the CB parameters are lifted, leading to a shift of the background R-symmetry CS level

$$k_R = -\frac{N(2N+1)}{2}\mathrm{sgn}(d).\tag{106}$$

The instantonic non-mixed level is left unchanged by this operation, as well as all the levels involving the anti-symmetric symmetry, under which the gauginos are not charged. At sufficiently weak coupling, namely $1/g^2 \gg \sqrt{d}$, we expect the full anti-symmetric hypermultiplet to remain massless under this deformation. The low energy theory is then expected (at least at weak coupling) to be a non-SUSY $Sp(N)$ YM theory coupled to an anti-symmetric hypermultiplet. The resulting phase diagram is shown in figure (6), where as usual we employ the $\mathbb{Z}_2 \times \mathbb{Z}_2$ Weyl symmetry at the fixed point to relate the four quadrants and the corresponding CS levels. We see then that on the $d$ axis, the non-mixed levels jump by the following amount

$$\Delta\mathrm{CS}_I = 4N\,, \quad \Delta\mathrm{CS}_R = N(2N+1)\,,\tag{107}$$

indicating the presence of a phase transition analogous to the $E_1$ case.

Before closing this section, it is worth commenting on the geometric realization of the previous SUSY breaking deformations, since it can give us indications on the stability of the theory. The methods we employed above lose their validity when we go to a sufficiently large coupling. To have access to this regime, we can nevertheless employ the $(p,q)$ web description of the theory, which in the $E_1$ theory context was able to describe consistently both the supersymmetric and the non-SUSY deformations involving the current multiplet [23]. In the case at hand, we have multiple possible SUSY preserving and SUSY breaking deformations, so the geometric description becomes more involved. In the following, we first consider the non-SUSY deformations of the $Sp(N)+1\mathbf{AS}$ theory at weak coupling, in the regime of validity of the field theory analysis, and then comment on the deformations at strong coupling.

The CFT point enjoys an SO(3)$_I$×SU(2)$_A$ global symmetry. To the two groups, we associate two independent supersymmetric mass deformations, $h$ and $m$. We can first turn on $h$ without introducing $m$. In this way, the web opens as in figure (108). The symmetry SU(2)$_A$ remains preserved, since the 7-branes on which the global symmetry is hosted (A and B in the figure) are aligned on the same 5-brane prong direction.

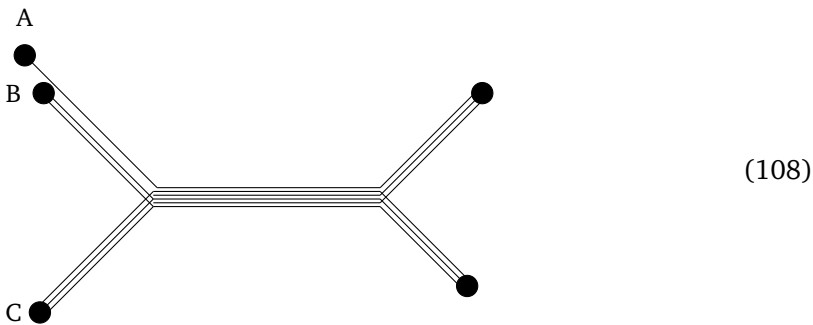

(108)

We can also break SU(2)$_A$ without breaking the instantonic SO(3)$_I$ symmetry, separating the two 7-branes A and B by introducing a mass $m \neq 0$. We end up with the $(p,q)$ web of figure (109). The field theory hosted by the web change as we vary $N$ [46]. The instantonic symmetry remains preserved, as it is hosted on the worldvolume of the two $[1,1]$ 7-branes, which remain parallel after the deformation.

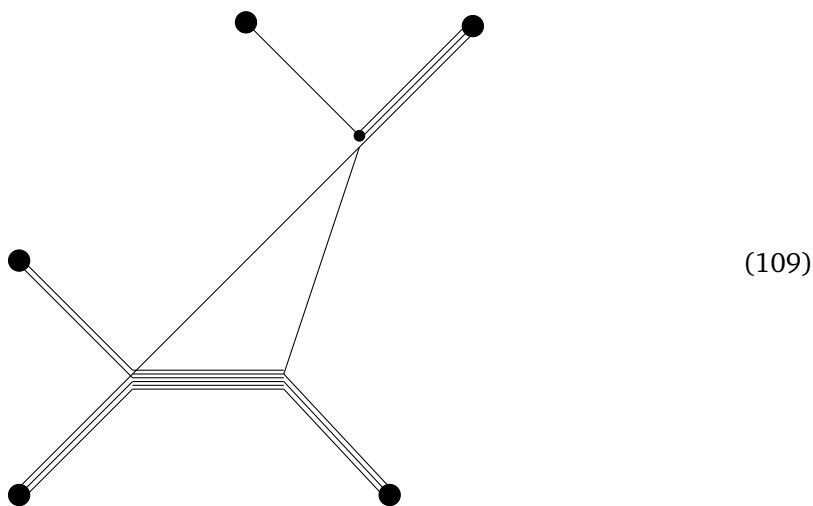

(109)

Finally, we can turn on both deformations, breaking the symmetry of the fixed point to U(1)$_I$×U(1)$_A$, as shown in figure (110).

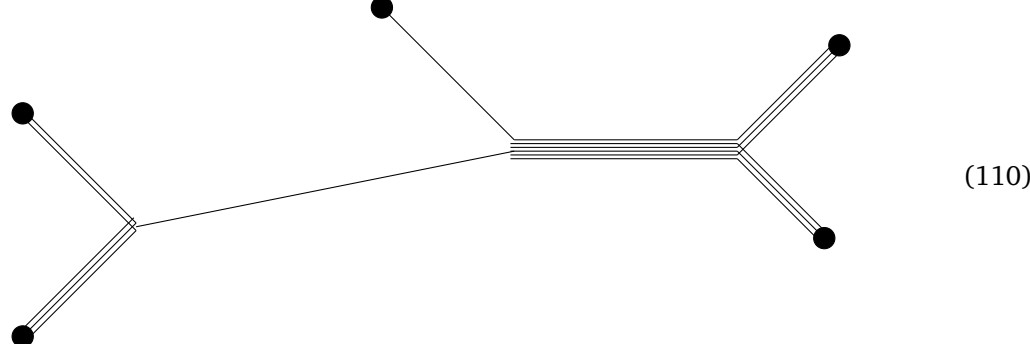

(110)

In all these cases, the breaking of the symmetry is visible by looking at the distance between the 7-branes which realizes the $\mathfrak{su}(2)_I$ and the $\mathfrak{su}(2)_A$ algebras. When the branes are separated,

strings stretching between the branes become massive, and the $\mathfrak{su}(2)$ theory hosted on their worldvolume is Higgsed down to U(1).

Similarly, non-SUSY deformations are associated with geometric moves of the brane web. Let us denote the plane on which the $(p,q)$-web lies as the $(x,y)$ plane, and a Dirichlet direction for all the 5-branes of the web as $z$. As pointed out in [23], since these deformations break the R-symmetry and part of the global symmetry by giving a VEV to a D-term, they can be realized as rotations of some 5-brane along the $x$-$z$ plane.[15] At the fixed point, giving a VEV to the D term of the flavor symmetry, the global symmetry breaks to $SU(2)_{AS} \to U(1)_A$ and the R-symmetry breaks down to $SO(2)_R$. Similarly, if we introduce a VEV for $SO(3)_I$, we break it to $U(1)_I$ together with the R-symmetry, while $SU(2)_A$ remains preserved.

If we go at weak coupling, only a Cartan subgroup $U(1)_I$ remains preserved by the supersymmetric deformation $h$. We then have three options: we can turn on a D-term for the flavor symmetry, which gets broken to $U(1)_A$, a D-term for the instantonic symmetry, which does not further break the instantonic symmetry, and a D term for both. Note that, in order to perform the deformation also in the weak coupling regime, we cannot completely break the instantonic symmetry turning on a generic D term VEV at the fixed point. Instead, we should tune the direction of this VEV in order to be compatible with the direction preserved by the SUSY deformation itself [21].

A D-term VEV for the gauge theory living on the worldvolume of a couple of 7-branes hosting an $\mathfrak{su}(2)$ algebra is realized by rotating one brane with respect to the other by a non-supersymmetric angle $\alpha$ [23]. Indeed, this operation breaks both the gauge symmetry on their worlvolume (to a $\mathfrak{u}(1)$ subalgebra) and the R-symmetry (down to $SO(2)_R$). We can then describe the geometric realization of the SUSY breaking deformations. Introducing a VEV for the D-term of $SU(2)_A$ is equivalent to a rotation around the D5 brane direction of the 5-prong associated with the 7-brane A, see figure (111).

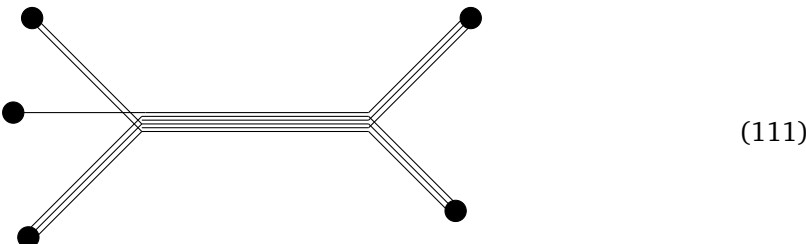

(111)

Indeed, looking at the 7-branes realizing the $SU(2)_A$ flavor symmetry, the deformation induces a D-term breaking on their worldvolume theory, breaking $SU(2)_A$ to $U(1)_A$. Moreover, at infinite coupling, we see that the deformation preserves the $SO(3)_I$ global symmetry enjoyed by the fixed point.

Note that the $(p,q)$ web suffers an instability along the direction of the Higgs branch parametrized by the breaking of the $(1,-1)$ 5-brane along the 7-brane B, even at finite coupling,[16] due to the non-supersymmetric rotation. This instability can be overcome by giving a supersymmetric mass $m$ to the anti-symmetric, leading to stabilization. All these features confirm the field theory analysis performed at weak coupling in section 2.

We can also introduce a VEV only for the D-term of $SO(3)_I$. This deformation is equivalent to

---

[15]Note that assuming the rotation plane to be $x$-$z$ plane for all branes is necessary to preserve the same $SO(2)_R$ subgroup of the original R-symmetry.

[16]This is known to be parametrized by the hypermultiplet in the trace of the anti-symmetric representation of $Sp(N)$, which is reducible [48].

a rotation of the whole three-junction shown in figure (112).

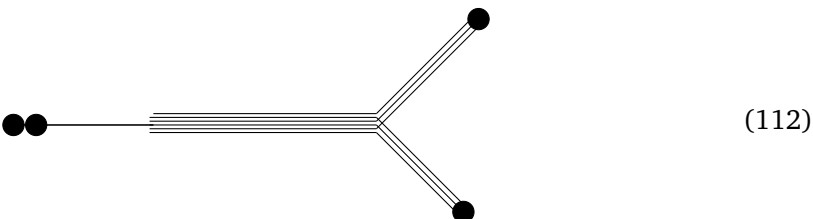

$$(112)$$

In this way, we preserve SU(2)$_A$ and break the instantonic symmetry when we rotated the branes at the fixed point, as expected. This is equivalent to a D-term on the worldvolume of the 7-branes describing the $\mathfrak{su}(2)_I$ algebra. The instability close to the fixed point comes from the tachyonic mass of the strings connecting the $[1,1]$ 7-branes. However, this can be overcome in the large $h$ limit, as happens to the $E_1$ case [23]. This reproduces all the features observed for this deformation in the field theory analysis of the previous section. At weak coupling, where the instability is resolved, the theory flows in the IR to Sp($N$) YM theory coupled to a full anti-symmetric hypermultiplet.

Finally, we can rotate the junction, leaving the brane A on the $x$-$y$ plane, figure (113). This breaks both SO(3)$_I$ and the SU(2)$_A$ global symmetries, leading to the two instabilities we found above, which can be resolved only at large $h$ and $m$.

This analysis shows a nice one-to-one correspondence between the non-supersymmetric deformations associated with the flavor symmetry of the UV fixed point and the expected non-supersymmetric deformations of the $(p,q)$ web. The field theory analysis at weak coupling is compatible with the $(p,q)$ web description, which in addition shows a series of instabilities at strong coupling invisible in field theory.

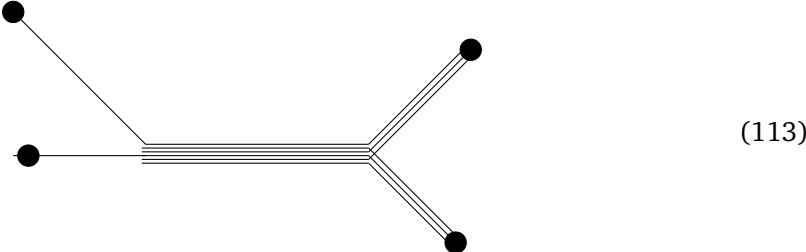

$$(113)$$

**Consistency with 't Hooft anomaly matching**

The theory has the same mixed 't Hooft anomaly between U(1)$_I$ and $\mathbb{Z}_2^{(1)}$ of the Sp($N$)$_{(N-1)\pi}$ theory, since inclusion of the anti-symmetric matter does not break the related symmetries. Therefore, for odd $N$, the partition function with the background gauge fields are changed under the minimal large gauge transformation as [22, 38]

$$Z_{\text{Sp}(N)+\text{AS}}[A, B] \rightarrow Z_{\text{Sp}(N)+\text{AS}}[A, B] \exp\left( \frac{2\pi i N}{4} \int_{\Sigma_4} \mathcal{P}(B) \right). \tag{114}$$

This again implies that the phase for odd $N$ cannot be trivial along RG flow and the proposed phase diagram in figure (6) is consistent with that.

## 5 SU(4)$_0$+2AS

Having analyzed the phase diagram of the $E_1^{(N)}$ SCFT, it is natural to generalize the previous analysis to the SU($2N$)$_0$+2**AS** SCFT point. These theories reduce to Sp($N$)+1**AS** if we go on

a point of their Higgs branch and to $SU(2N-1)_{2N-1}$ when we go at infinite distance on one of the CB directions. For these reasons, these theories can be considered as a sort of "mother theory" for many of the cases we analyzed above. At weak coupling, they enjoy a $U(2)_A \times U(1)_I$ global symmetry, that gets enhanced to $U(2) \times SO(3)_I$ at the fixed point [49]. The inverse gauge coupling squared $h$ is associated with the global symmetry, while the two independent masses $m_T, m$ are respectively the masses associated with the $U(1)$ and the $SU(2)$ part of the $U(2)$ flavor group. In the following, we will study the case $N = 2$. This is particularly interesting, since $\mathfrak{su}(4) \cong \mathfrak{so}(6)$ and this will have consequences on the global symmetry of the theory. In the generic $N$ case, it is still not completely clear how we can reach the CFT phase from the perturbative one. Therefore, we leave the discussion on this generalisation for a future analysis.

For $N = 2$, the gauge symmetry is $SU(4) \cong \mathrm{Spin}(6)$, so the global symmetry is $U(1)_I \times Sp(2)/\mathbb{Z}_2$, since the anti-symmetric of $SU(4)$ is the vector of $SO(6)$, which has an $Sp(2)$ global symmetry. This can be directly seen from the perturbative prepotential

$$\mathcal{F} = h\sum_{i=1}^{4} a_i^2 + \frac{1}{6}\sum_{i<j}(a_i - a_j)^3 - \frac{1}{12}\sum_{i<j}|a_i + a_j + m_T \pm m|^3\,, \tag{115}$$

where the two anti-symmetric of $SU(4)$ can be rewritten as two vectors of $SO(6)$ under the identification

$$a_1 = \frac{1}{2}(\phi_1 + \phi_2 - \phi_3)\,, \quad a_2 = \frac{1}{2}(\phi_1 - \phi_2 + \phi_3)\,, \quad a_3 = \frac{1}{2}(-\phi_1 + \phi_2 + \phi_3)\,, \tag{116}$$

where $\phi_i$ are independent variables obeying the relations $\phi_1 \geq \phi_2 \geq \phi_3 \geq 0$ in our chosen Weyl chamber. In this basis, the W bosons of $SU(4)$ can be rewritten as

$$\sum_{i<j}^{4}(a_i - a_j)^3 = \sum_{i<j}^{3}(\phi_i - \phi_j)^3 + \sum_{i<j}^{3}(\phi_i + \phi_j)^3\,, \tag{117}$$

while a (massless) anti-symmetric hypermultiplet as

$$\sum_{i<j}^{3}(a_i + a_j) = \sum_i \phi_i\,, \quad \sum_{i=1}^{3}(a_i + a_4) = -\sum_i \phi_i\,, \tag{118}$$

so the expression in eq. (115) becomes

$$\mathcal{F} = h\sum_{i=1}^{3}\phi_i^2 + \frac{1}{6}\sum_{i<j}^{3}\left[(\phi_i - \phi_j)^3 + (\phi_i + \phi_j)^3\right] - \frac{1}{12}\sum_{i=1}^{3}|\phi_i \pm m_T \pm m|^3\,, \tag{119}$$

which is nothing but the prepotential of $SO(6)$ coupled to two vector hypermultiplets.

The action of the Weyl group on the product $w \cdot \vec{m}^{\mathrm{Sp}(2)}$ can be inferred from the expression of the simple roots in terms of the orthonormal basis $\{e_1, e_2\}$

$$\alpha_1 = e_1 - e_2\,, \quad \alpha_2 = 2e_2\,. \tag{120}$$

An element $z_i$ of the Weyl group acts on a weight $w$ as $w \to w - (w \cdot \alpha_i^\vee)\alpha_i$, so it can be converted into an action on $\vec{m}$ as [32]

$$z_1: \ m \leftrightarrow m_T\,, \quad z_2: \ m_T \to -m_T\,.$$

Combining together, we obtain

$$z_1': \ m \to -m\,, \quad z_2: \ m_T \to -m_T\,.$$

so we see that both $m$ and $m_T$ transform under the Weyl action.

The correct parametrization for the brane web is shown in figure (121), where $m_1 = m + m_T$, $m_2 = m_T - m$.

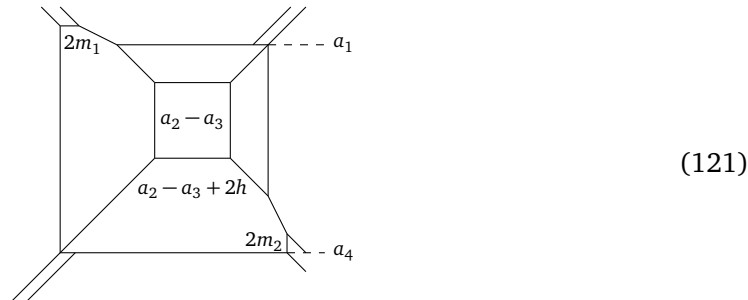

(121)

Notice that in this phase we cannot go on the Higgs branch of the theory by breaking the $(1, -1)$ brane in the low-right part of the web into each other, since this would violate the s-rule. As a consequence, we cannot obtain the Sp(2)+1AS analyzed in section 4. In order to do so, we need to flop two hypermultiplets $a_2 + a_3 \pm m - m_T$ and reach the phase shown in figure (122).

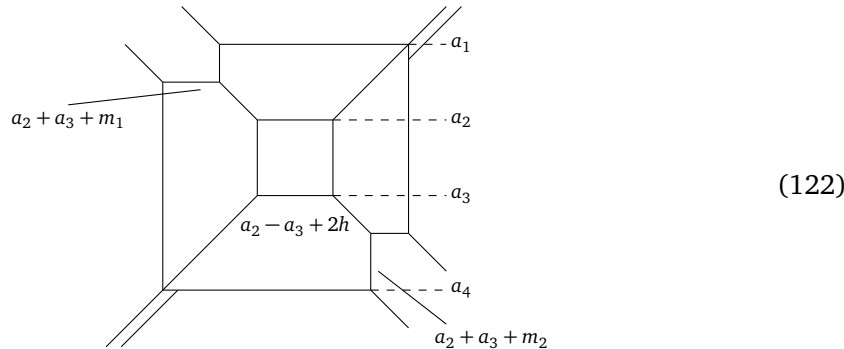

(122)

In this phase, we can select a specific point of the CB $a_2 = -a_3$ and switch off the mass parameter $m = m_T$. At this particular point, two hypermultiplets $a_1 + a_4 - m + m_T$ and $a_2 + a_3 - m + m_T$ become massless and we can enter the Higgs branch by breaking one of the $(1, -1)$ 5-brane with respect to the 7-brane attached to the other. The $(p, q)$ web reduces then to the Sp(2) one in figure (79) under the identification $a_i^{\text{Sp}} = a_i^{\text{SU}}$, $i = \{1, 2\}$ and $m^{\text{Sp}} = 2m^{\text{SU}}$. As we will see later, also the complete prepotential will coincide (up to some subtlety).

Under S-duality, we obtain the duality map

$$\bar{a}_1 = a_1 + h, \quad \bar{a}_2 = a_2 + h, \quad \bar{a}_3 = a_3 - h, \quad \bar{h} = -h, \quad m_T = \bar{m}_T, \quad m = -\bar{m}. \quad (123)$$

We can obtain the CS terms by looking directly at the duality map. The perturbative prepotential can be augmented by three additional CS terms for the background fields

$$\mathcal{F} = h \sum_{i=1}^{4} a_i^2 + \frac{1}{6} \sum_{i<j} (a_i - a_j)^3 - \frac{1}{12} \sum_{i<j} |a_i + a_j + m_T \pm m|^3 + \alpha h^3 + \beta h m^2 + \gamma h m_T^2, \quad (124)$$

and this should remain invariant under the duality map. This invariance is guaranteed if and only if

$$\left(\alpha + \frac{2}{3}\right) h^2 + (\beta + 1) h m^2 + (\gamma + 1) h m_T^2 = 0, \quad (125)$$

which is solved by $\alpha = -\frac{2}{3}$, $\beta = \gamma = -1$. This gives us the CS terms

$$c_{hhh} = c_{hmm} = c_{hm_T m_T} = -4. \quad (126)$$

Alternatively, we can obtain the same result by constructing the CFT prepotential. It is a simple matter to construct the invariant CB parameters

$$\hat{a}_i = a_i + \frac{h}{2}, \quad \hat{a}_3 = a_3 - \frac{h}{2}. \tag{127}$$

Substituting these invariant CB parameters in the expression for the gauge couplings, we obtain

$$(g^{-2})_{ii} = (6-2i)\hat{a}_i + 2\sum_{k=1}^{i-1}\hat{a}_k + 4\hat{a}_T,$$

$$(g^{-2})_{i<j} = 6\hat{a}_T - 2\hat{a}_i.$$

The CFT prepotential reads then

$$\mathcal{F}_{CFT} = \frac{1}{6}\sum_{i<j}(\hat{a}_i - \hat{a}_j)^3 - \frac{1}{6}\sum_{i<j}|\hat{a}_i + \hat{a}_j|^3 - h^2(\hat{a}_1 + \hat{a}_2) + 2(m^2 + m_T^2)\hat{a}_4. \tag{128}$$

Expanding the CFT prepotential, we consistently obtain the correct CS terms for the global symmetries. This prepotential reduces to the Sp(2) +1AS one when we go on the Higgs branch of the theory. In order to see this, we first flop the hypermultiplets of mass

$$\hat{a}_1 + \hat{a}_4 + m_T \pm m, \tag{129}$$

in order to reach the phase in figure (122). Then, we set $m_T = m$ and $a_2 = -a_3$. The invariant CB parameters look the same if we identify $a_1^{Sp} = a_1^{SU}$, $a_2^{Sp} = a_2^{SU}$ and $h^{Sp} = h^{SU}$ as expected, since in the operation the SO(3)$_I$ global symmetry remains preserved. The CFT prepotential changes as

$$\mathcal{F} = \frac{1}{6}\sum_{i<j}(\hat{a}_i - \hat{a}_j)^3 - \frac{1}{6}\sum_{i<j}|\hat{a}_i + \hat{a}_j|^3 - h^2(\hat{a}_1 + \hat{a}_2) + 2(m^2 + m_T^2)\hat{a}_4 - \frac{1}{6}(\hat{a}_1 + \hat{a}_4 + m_T \pm m)^3. \tag{130}$$

In the limit $a_2 = -a_3$ and $m = m_T = m^{Sp}/2$ this reduces precisely to the Sp(2)+1AS prepotential (where we dropped the Sp label)

$$\mathcal{F}_{CFT}^{Sp(2)} = \frac{4}{3}(\hat{a}_1^3 + \hat{a}_2^3) - m^2\hat{a}_1 - h^2(\hat{a}_1 + \hat{a}_2) - \frac{1}{6}m^3, \tag{131}$$

up to a cubic CS term in $m$.

Let us briefly comment on this additional term. Its presence is related to the non-invariance of eq. (130) under the Weyl group reflection associated with $m_T$, since only a subset of the entire orbit of the Weyl group was flopped to go from the CFT to the Sp(2) phase. However, since $m = m_T$, this translates into a non-invariance w.r.t. the Weyl group associated with $m$ in the Sp(2) theory. In order to respect the Weyl symmetry, this should be then eliminated, reducing the prepotential to the expected one. Notice that we are also able to reproduce the perturbative hypermultiplets of the Sp(2) theory, starting from the perturbative hypermultiplets of SU(4). These read

$$a_i + a_j \pm m + m_T, \tag{132}$$

and in the Sp(2) limit reduce to (here we omit the massless ones)

$$m, \quad a_1 \pm a_2 + m, \quad a_1 \pm a_2, \quad -(a_1 \pm a_2 - m), \quad -(a_1 \pm a_2). \tag{133}$$

These are nothing but the perturbative hypermultiplets of Sp(2), namely $a_1 \pm a_2 \pm m$, together with the additional W bosons necessary in order to reproduce the correct contribution for Sp(2).

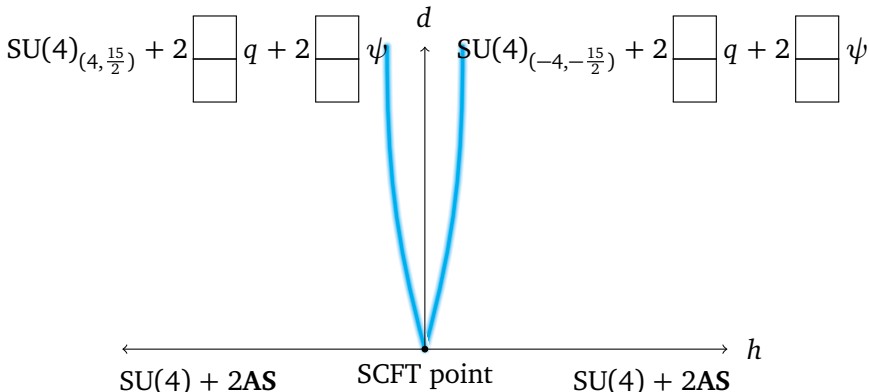

Figure 7: Summary of the phase diagram for the $SU(4) + 2\mathbf{AS}$ case.

Since the global symmetry has rank 3, we expect three independent mass parameters associated with the non-SUSY global current deformations of the fixed point. In order to ensure the stability of the weakly coupled description after the deformation, we choose to give a VEV to a D-term for the instantonic current multiplet. This breaks $SO(3)_I$ to $U(1)_I$ as well as $SU(2)_R$ to $U(1)_R$, while keeping the full flavor symmetry preserved. In the weakly coupled description, this gives a mass to both the gauginos and the scalar gauginos, while keeping the anti-symmetric hypermultiplets fully massless (at least at tree level). In the decoupling, the R-symmetry CS level gets shifted as

$$k_R = -\frac{15}{2}\text{sgn}(d), \tag{134}$$

leading to the phase diagram shown in figure (7). The CS levels shift going from $h > 0$ to $h < 0$ as

$$\Delta c_{hhh} = \Delta c_{hmm} = \Delta c_{hm_Tm_T} = 8, \quad \Delta k_R = 15, \tag{135}$$

indicating a phase transition around the non-supersymmetric deformation axis.

**Consistency with 't Hooft anomaly matching**

The inclusion of an anti-symmetric to the pure $SU(2N)_0$ theory breaks the one-form symmetry $\mathbb{Z}_{2N}^{(1)}$ to $\mathbb{Z}_2^{(1)}$. Therefore the theory has the same structure on the mixed 't Hooft anomaly between $U(1)_I$ and $\mathbb{Z}_2^{(1)}$ as the $SU(N+1)_{N+3}$ theory with odd $N$ we analyzed in previous section. Again, under a large gauge transformation with unit winding, the partition function changes as [22, 38]

$$Z_{SU(2N)_0+2\mathbf{AS}}[A, B] \rightarrow Z_{SU(2N)_0+2\mathbf{AS}}[A, B] \exp\left(\frac{2\pi i N(2N-1)}{4} \int_{\Sigma_4} \mathcal{P}(B)\right), \tag{136}$$

which indicates that the 't Hooft anomaly is nontrivial for odd $N$ and trivial for even $N$. Therefore, the phase for odd $N$ cannot be trivial along RG flow, while this argument does not give any constraint for the $SU(4)$ case shown in figure (7).

## 6 Conclusions and discussion

In this paper, we studied supersymmetry breaking deformations of 5d SYM theories, as well as theories coupled to matter hypermultiplets, and showed the existence of phase transitions

separating different weakly coupled regions of their phase diagram. That brane systems can be useful for the study of non-supersymmetric Yang-Mills theory was noted long ago in [50]. Recently, there has been a flurry of results about dualities and dynamics of 3d gauge theories without supersymmetry using brane systems, see for instance [24, 25, 37, 51]. In higher dimensions, and especially in $d \geq 5$ there are far fewer studies, with some notable exceptions like [23]. It is therefore well overdue to explore the consequences of brane dynamics for non-SUSY QFTs in 5d. Our study has revealed that 5-brane webs can also be useful in understanding the strong coupling dynamics of non-supersymmetric deformations of 5d SCFTs.

One of the main motivations for our work was the question of whether 5d fixed points without supersymmetry can exist in spacetime dimensions $d > 4$. It is not easy to provide a definitive answer to this question yet. In particular, while we can conclusively argue for the existence of phase transitions, the order of such transitions is not straightforward to determine. It would obviously be interesting to find further arguments in clarifying this question. Careful analysis of symmetries and anomalies are an indispensable tool in constraining the space of possible scenarios. It would, in particular, be interesting to expand on the discussion in appendix A by turning on further backgrounds for the 0-form flavor symmetries of the theories with matter hypermultiplets and look for discrete anomalies along the lines of [52].

Among the various theories we analyzed above, the presence of instability on the Higgs branch when the non-SUSY deformation was turned on at infinite coupling was a common feature, with however a notable exception. Indeed, the UV fixed point associated with the $SU(N + 1)_{N+3}$–$Sp(N)_{(N+1)\pi}$ theories has no Higgs branch at the fixed point for $N$ even and, as a consequence, no instabilities of this kind can arise. This can be an example of an RG flow between a SUSY and a non-supersymmetric CFT purely at strong coupling. However, we cannot exclude additional instabilities arising through other mechanisms. These can be, in principle, analyzed by looking at the string theory construction of these theories. This can be a possible interesting direction to pursue in the future.

In the case of holographic supersymmetric fixed points, much of the insight into their dynamics is obtained through their dual $AdS_6$ geometries [5, 53, 54]. Recently, non-SUSY $AdS_6$ solutions of type II supergravities have appeared in [55, 56]. Moreover the aforementioned class of solutions are close cousins of the SUSY solutions dual to 5d SCFTs. It would be interesting to explore potential similarities between our field theory deformations, and these studies. In particular while the effective gauge theoretic description breaks down close to the transition lines outlined in this paper, the supergravity description might provide a window into the dynamics of this regime. Another particularly relevant solution is the massive IIA solutions of Brandhuber and Oz [7], whose dual SCFT is the rank-$N$ $E_1$ SCFT. We have performed a detailed analysis of the SUSY-breaking deformation of this theory, identifying their geometric realization in the 5-brane web. It would be interesting to identify the corresponding deformation in the holographic setup.

Last but not least, generalizing and classifying SUSY breaking deformations might shed light on understanding UV behaviors of phenomenological models with extra dimensions. While such models are always perturbatively non-renormalizable, in many cases it is unclear whether or not they admit a UV completion and, if so, how they behave at high energies. Pursuing the direction of our work, one can try to identify UV completions and their properties of some phenomenological models.

## Acknowledgments

We thank Adi Armoni, Oren Bergman, Amihay Hanany, Hirotaka Hayashi, Sung-Soo Kim, Carlos Nunez, Tomohiro Shigemura, Ricardo Stuardo, and Shigeki Sugimoto for useful discussions and/or comments on the draft.

**Funding information** M.A. is supported by a JSPS postdoctoral fellowship grant No. 22F22781. M. H. is supported by MEXT Q-LEAP, JST PRESTO Grant Number JPMJPR2117, JSPS Grant-in-Aid for Transformative Research Areas (A) JP21H05190 and JSPS KAKENHI Grant Number 22H01222. F.M. is partially supported by the Israel Science Foundation under grant No. 1254/22.

# A Details of the anomaly computation

In this appendix we collect some results which are useful in computing the anomalies stated in the main text. The following material is by now standard and appears in many papers, but we find it convenient to collect these results in order to be self-contained. We closely follow the logic and presentation of [52]. In order to describe $SU(N)/\mathbb{Z}_k$ gauge theory on a spacetime manifold $X$, we first embed it into a larger group [57–59]

$$G = \frac{SU(N) \times U(1)}{\mathbb{Z}_k}. \tag{A.1}$$

A $G$-bundle is specified by the one-forms $A_i$ defined on open covers $U_i$, together with their transition functions $t_{ij}$. On double overlaps $U_{ij} = U_i \cap U_j$ the gauge fields are related by

$$A_j = t_{ij}^{-1} A_i t_{ij} + t_{ij}^{-1} d t_{ij}. \tag{A.2}$$

On triple overlaps $U_{ijk} = U_i \cap U_j \cap U_k$, the transition functions must satisfy the cocycle condition

$$t_{ij} t_{jk} t_{ki} = 1. \tag{A.3}$$

Let us write the $G$-connection $A$ in terms of an $\mathfrak{su}(N)$ connection $\tilde{A}$ and a $\mathfrak{u}(1)$ connection $\hat{A}$ via

$$A = \tilde{A} + \frac{1}{k}\hat{A}\mathbb{1}. \tag{A.4}$$

Now the cocycle condition in eq. (A.3) can be satisfied for transition functions $\tilde{t}_{ij}$ and $\hat{t}_{ij}$ obeying

$$\tilde{t}_{ij} \tilde{t}_{jk} \tilde{t}_{ki} = \exp\left(\frac{2\pi i}{k} n_{ijk}\right), \qquad \hat{t}_{ij} \hat{t}_{jk} \hat{t}_{ki} = \exp\left(-\frac{2\pi i}{k} n_{ijk}\right). \tag{A.5}$$

Note the redundancy in redefining the $\mathfrak{su}(N)$ transition functions by an arbitrary phase

$$\tilde{t}_{ij} \to \tilde{t}_{ij} \exp\left(\frac{2\pi i}{k} n_{ij}\right). \tag{A.6}$$

This defines a relation

$$n_{ijk} \sim n_{ijk} + n_{ij} + n_{jk} + n_{ki}. \tag{A.7}$$

Modding out by this relation is equivalent to introducing a 2-form gauge field $B \in H^2(X, \mathbb{Z}_k)$.

A continuum description of a discrete 2-form gauge theory is obtained by imposing the following constraint

$$kB = d\hat{A}, \tag{A.8}$$

and requiring invariance of the system under the following 1-form gauge symmetry

$$B \to B + d\lambda, \qquad \hat{A} \to \hat{A} + k\lambda, \tag{A.9}$$

where $\lambda$ is an ordinary (1-form) gauge field. The above gauge transformation removes the extra U(1) symmetry we started out with so that we end up with a $G/U(1) \cong SU(N)/\mathbb{Z}_k$

bundle. The crucial point now is that the instanton current for the original U($N$) field strength $F = dA + A^2$ is not invariant under this 1-form gauge symmetry. To make it gauge invariant we send

$$F \to F - B\mathbb{1}, \tag{A.10}$$

at the expense that the instanton current is no longer integer quantized

$$\frac{1}{8\pi^2} \int_{\Sigma_4} \text{tr}\left[(F-B)\wedge(F-B)\right] = \frac{1}{8\pi^2} \int_{\Sigma_4} \left[\text{tr}(F\wedge F) - 2\text{tr}(F)\wedge B + NB\wedge B\right]. \tag{A.11}$$

When $B$ is dynamical, this describes the $SU(N)/\mathbb{Z}_k$ gauge theory, while if $B$ is a fixed background it is regarded as an $SU(N)$ gauge theory with a background gauge field of the $\mathbb{Z}_k^{(1)} \subseteq \mathbb{Z}_N^{(1)}$ one-form symmetry. Using the relation in eq. (A.8) and the quantisation condition

$$\frac{1}{8\pi^2} \int_{\Sigma_4} \left[\text{tr}(F\wedge F) - \text{tr}F \wedge \text{tr}F\right] \in \mathbb{Z}, \tag{A.12}$$

we can extract the fractional part of the instanton number

$$\frac{N(N-1)}{8\pi^2 k^2} \int_{\Sigma_4} d\hat{A} \wedge d\hat{A}. \tag{A.13}$$

The above discussion is useful to compute the mixed 't Hooft anomaly between the U(1)$_I$ and $\mathbb{Z}_k^{(1)} \subseteq \mathbb{Z}_N^{(1)}$ symmetry of the 5d SU($N$) gauge theory following [22, 38, 60]. After turning on the background gauge field $B$ for $\mathbb{Z}_k^{(1)}$, let us turn on a background gauge field $A_I$ for the U(1)$_I$ by adding the following term to the Lagrangian

$$\delta\mathcal{L} = iA_I \wedge \star J_I[F-B]. \tag{A.14}$$

Assuming our space time admits non-trivial 1-cycle $\gamma$, under a large gauge transformation winding $\gamma$, the gauge field transforms as $A_I \to A_I + \zeta$, where by definition, $\oint_\gamma \zeta/2\pi$ integrates to $\ell \in \mathbb{Z}$. Therefore under such a transformation, the partition function changes by

$$Z[A_I, B] \to Z[A_I, B]\exp\left(2\pi i\ell\frac{N(N-1)}{k^2}\right), \tag{A.15}$$

which is in perfect agreement with the anomalous phase computed in [38, 60].

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
