# Peer review of "d SCFTs and their non-supersymmetric cousins"

_SciPost Physics, doi:SciPost Phys. 16, 036 (2024)_

## Round 2 · Referee Report · Anonymous (Referee 1) · 2023-10-9

Strengths

The paper studies supersymmetry-breaking deformations of five-dimensional superconformal field theories. Combining the analysis of the 5d prepotential of ref [30] with the methods developed in references [20] and [21], the authors are able to study numerous families of theories. Of particular interest is the case of Sp(N) SYM with an antisymmetric hyper, which has a known holographic dual [46]. The study of the deformation from field theory thus highlights the importance of the study of non-SUSY deformations in gravity.

Weaknesses

Some presentation issues in the introduction and minor typos, but easily solvable with minor revisions.

Report

The paper uses field-theoretic and geometric methods to investigate the existence of non-supersymmetric conformal field theories in five dimensions. This is argued via a chain of arguments: first one studies deformations of the SCFT leading to SYM phases, which are described by the prepotential. The authors stress the importance of integration constants in the prepotential, which represent Chern-Simons levels for background vector multiplets. When different deformations are related by a symmetry of the fixed point, the gauge theory phases are related as well, and are sometimes referred to as ``UV duals''. Focusing on these, the authors are able to fix the Chern-Simons levels for the background vector multiplets. One then deforms breaking supersymmetry and, in presence of a particular change of levels, argue in favour of the existence of a phase transitions and a non-supersymmetric fixed point, of which the authors study the stability as well.

The paper represents a significant contribution to the literature, and I would recommend it for publication after the authors address the minor revisions described below.

Requested changes

1) Presentation. One of the key concepts used in the paper is that of "UV duality". This is something of a misnomer, accepted because of its common usage in the literature. I believe that it should be emphasized in the introduction that "UV dualities" between apparently inequivalent gauge theories are simply due to related susy-preserving mass deformations. This point is already present in the article at p. 6, but to avoid imprecisions I believe that it would be better to bring this forward to the introduction, taking the place of the sentence about the "continuation past infinite coupling of the weakly coupled gauge theory description" (p. 1 and p. 3). The RG flow is unidirectional, and the "infinite coupling limit" is not a well-defined operation, as stressed, for instance in [1812.10451].

2) Typos and clarifications. - p. 1 +4: Is there a reason why ref [46] is not grouped together with [4-6] in the introduction? - p. 1 +14: that the global symmetry of $E_1$ is $SO(3)_I$ instead of $SU(2)_I$ was first suggested in [36] - p. 1 +17 (and elsewhere): it should be "Cartan subgroup" rather than "Cartan" - p. 1 -17: I believe that the sentence would be clearer as "is however related to the original weakly coupled description by" - p. 2 +2: "on a generic" $\to$ "at a generic" - p. 2 -2: "the background CS levels for the instantonic symmetry" - p. 6 - 18: Could the authors please clarify, perhaps in a footnote, what they mean by a "non-perturbative hypermultiplet"? - p. 8 +5: "in terms" - p. 9 +12: that the global symmetry of the fixed point of $SU(N)_N$ is $SO(3)_I$ instead of $SU(2)_I$ was first suggested in [31] - p. 11 (33): I believe that it would be clearer if $\alpha$ was introduced in the text, e.g. "this integration constant $\alpha$ into the IMS prepotential" - p. 12 +4: is there a proof of the fact that the Weyl group of the global symmetry of the SCFT should be S-duality in the $(p,q)$ web, apart from the fact that they have a related action? If not, perhaps "implemented" is too strong compared to "related"? - p. 14 +4: "the gauginos being in the adjoint" - p. 14 footnote 5: what group are the authors referring to? $d_{abc}$ for $SU(N)$ is not vanishing for $N\geq 3$. - p. 15 - 3: "a shift of the CS level" - p. 28 -8: "reach" - p. 32 -8: I suppose that the authors here restricting to holographic supersymmetric fixed points?

  • validity: -
  • significance: -
  • originality: -
  • clarity: -
  • formatting: -
  • grammar: -

Author:  Mohammad Akhond  on 2023-10-20  [id 4048]

(in reply to Report 1 on 2023-10-09)

We thank the referee for their comments on our paper. We have implemented all the changes requested. Please see below for detailed breakdown.

1) We have rephrased the paragraph in the opening page to stress the direction of the RG and clarify the precise meaning of UV-duality

2) The typos have been corrected as follows

  • P.1+4: References are now grouped together.
  • p. 1 +14: Indeed, we should have been more careful to give credit, the reference now appears in footnote 1
  • p. 1 +17 (and elsewhere): Fixed
  • p. 1 -17: We agree, we have modified this sentence accordingly
  • p. 2 +2: Fixed
  • p. 2 -2: Fixed
  • p. 6 - 18: Thanks for pointing out the lack of clarity. We added footnote 4 to explain.
  • p. 8 +5: Fixed
  • p. 9 +12: Reference added
  • p. 11 (33): We agree, thanks for your suggestion. We changed the sentence accordingly.
  • p. 12 +4: We are not aware of a rigorous proof, other than the observation that it works in most cases. We changed the sentence to reflect this.
  • p. 14 +4: Fixed
  • p. 14 footnote 5: We are referring to $d_{abc}$ in the adjoint representation. While for arbitrary irrep of SU(N) it is non-vanishing it vanishes for the adjoint. Point here being that integrating out the gauging (which is adjoint valued) does not modify the CS level for the gauge field. We modified the sentence to make this more clear.
  • p. 15 - 3: Fixed
  • p. 28 -8: Fixed
  • p. 32 -8: Indeed we are referring to holographic fixed points. We changed the wording in that sentence to reflect this more clearly

---

## Round 4 · Referee Report · Anonymous (Referee 1) · 2023-10-23

Report

I would like to thank the authors for answering my questions and implementing the changes.
I recommend the paper for publication.

---

## Round 4 · Referee Report · Anonymous (Referee 2) · 2023-12-2

Strengths

This paper analyzes whether non-supersymmetric deformations of five-dimensional superconformal field theories with a rank greater than one can give rise to an interacting 5D fixed point without supersymmetry. This generalizes previous results obtained in the rank-one case. The use of $(p,q)$-fivebrane webs plays a prominent role in this analysis.

Weaknesses

Understanding the dynamics of 5D SCFTs and their non-supersymmetric deformations is a challenging task. The $(p,q)$-brane webs approach certainly offers useful consistency checks, but it is not completely clear whether it can also provide definitive answers to the questions that the authors would like to address.

Section 2.2 has a a couple of presentation issues , but they can be easily resolved with some minor revisions.

Report

The paper makes a worthy contribution to the existing literature on 5D SCFTs, and I recommend it for publication once the authors address the minor revisions outlined below.

Requested changes

1) On page 6, the authors refer to "non-perturbative hypermultiplets." Please see additional comments in Footnote 4. I suggest removing this terminology, the footnote and any subsequent references to it, as it holds little significance. Additionally, contrary to what is stated in Footnote 4, there is no QFT proof that 5D supersymmetric gauge theories or SCFTs exhibit S-duality.

2) Overall, Section 2.2 is unnecessarily verbose, containing numerous small subsections that could be easily integrated into a single section. To enhance the reader's experience, it would be beneficial if the authors could provide a more streamlined version of Section 2.2 upon resubmission.

  • validity: -
  • significance: -
  • originality: -
  • clarity: -
  • formatting: -
  • grammar: -

Author:  Mohammad Akhond  on 2023-12-25  [id 4210]

(in reply to Report 2 on 2023-12-02)

I wish to thank the referee for their comments.

1) All reference to non-perturbative hypermultiplets is now removed, instead we use the terminology non-perturbative states, which is more accurate. Note, that v6 is the latest draft, somehow I missed one such phrase in v5 which is resubmitted here.

2) This section has now been streamlined along the lines suggested by the referee.

Thanks again for your comments.

---

## Editorial Decision

published